# MobileIPL: Enhancing Mobile Agents Thinking Process via Iterative Preference Learning

**Kun Huang**[1*], **Weikai Xu**[21*†], **Yuxuan Liu**[31†], **Quandong Wang**[1], **Pengzhi Gao**[1], **Wei Liu**[1], **Jian Luan**[1], **Bin Wang**[1], **Bo An**[2‡]
[1]Xiaomi Inc. [2]Nanyang Technological University
[3]Gaoling School of Artificial Intelligence, Renmin University of China
huangkun11@xiaomi.com, xuwk266@gmail.com

## Abstract

The Chain of Action-Planning Thoughts (CoaT) paradigm has been shown to improve the reasoning performance of VLM-based mobile agents in GUI tasks. However, the scarcity of diverse CoaT trajectories limits the expressiveness and generalization ability of such agents. While self-training is commonly employed to address data scarcity, existing approaches either overlook the correctness of intermediate reasoning steps or depend on expensive process-level annotations to construct process reward models (PRM). To address the above problems, we propose an **I**terative **P**reference **L**earning (*IPL*) that constructs a CoaT-tree through iterative sampling, scores leaf nodes using rule-based reward, and backpropagates feedback to derive Thinking-level Direct Preference Optimization (T-DPO) pairs. To prevent overfitting during warm-up supervised fine-tuning, we further introduce a three-stage instruction evolution, which leverages GPT-4o to generate diverse Q&A pairs based on real mobile UI screenshots, enhancing both generality and layout understanding[§]. Experiments on three standard Mobile GUI-agent benchmarks demonstrate that our agent *MobileIPL* outperforms strong baselines, including continual pretraining models such as OS-ATLAS and UI-TARS. It achieves state-of-the-art performance across three standard Mobile GUI-Agents benchmarks and shows strong generalization to out-of-domain scenarios.

## 1 Introduction

VLM-based mobile agents (Wang et al., 2023; Ding, 2024) have attracted considerable attention due to their ability to seamlessly interact with mobile graphical user interfaces (GUIs) and their potential to autonomously perform daily tasks. Since actions are not directly specified in user instructions, mobile agents benefit from generating intermediate thoughts aligned with the current GUI context. Recent work such as AITZ(Zhang et al., 2024b) has demonstrated that the Chain of Action-Planning Thoughts (CoaT) pattern—resembling the slow-thinking "System 2" process—is particularly effective in GUI domains.

However, directly applying supervised fine-tuning (SFT) on CoaT trajectories may cause overfitting, leading the model to be trapped in fixed reasoning patterns. To address this limitation, recent studies in the general domain have explored self-training strategies. These approaches typically utilize the correctness of the final answer in output as a reward signal to train the model(Luong et al., 2024). While effective in some contexts, relying solely on final answers overlooks the quality of intermediate reasoning steps, which can result in reward hacking and suboptimal reasoning processes. Some search-based approaches, such as ReST-MCTS (Xie et al., 2024), tackle this problem by learning a process reward model (PRM) to evaluate individual reasoning steps. However, these approaches often require large-scale manual annotation of intermediate steps (Guo et al., 2025a). This challenge is especially severe in the Mobile GUI Agent domain. Unlike text-based tasks in coding or math, GUI

---

[*]Equal contribution.
[†]Work done during the internship at XiaoMi.
[‡]Bo An is the corresponding author.
[§]https://huggingface.co/datasets/xwk123/MobileIPL-dataset

environments rely on real devices or simulators, making step-level reward annotation significantly more costly and labor-intensive.

To address these limitations, we propose an iterative sampling framework that constructs a CoaT-tree based on Monte Carlo Tree Search (MCTS). Instead of relying on a PRM, we score each reasoning step and construct thinking-level DPO (T-DPO) pairs without manual step annotation. Specifically, we perform multi-turn dialogue with a vision-language model (VLM) to incrementally build a CoaT-tree, where each node corresponds to a sampled response at a given reasoning step, conditioned on the dialogue history. This hierarchical structure captures diverse reasoning paths and facilitates fine-grained assessment of intermediate thoughts. We first assign rewards to the leaf node, and then propagate these signals backward through the CoaT-tree to earlier reasoning steps. Based on the resulting values, we construct thinking-level DPO pairs to help agents optimize both final actions and the overall quality of their reasoning.

To mitigate the lack of diversity after warm-up SFT, we adopt an instruction evolution strategy. Specifically, we generate diverse Q&A pairs grounded in real mobile UI screenshots from downstream training datasets. These Q&A pairs serve two purposes: (1) prevent agents from overfitting to static downstream instructions by introducing varied reasoning contexts, and (2) improve agents' understanding of UI layouts through visually grounded question-answering. We evaluate our approach on the CoaT dataset AITZ and long-horizon dataset AMEX, where it outperforms state-of-the-art GUI-agent continual pretraining agents such as OS-ATLAS (Wu et al., 2024) (+4.04%) and UI-TARS (Qin et al., 2025) (+3.54%). Furthermore, experiments on the AndroidControl dataset demonstrate the strong generalization capability of our method to unseen apps and instructions (tasks). Under limited training resources, IPL consistently outperforms naive DPO using only half of the data for one iterative training round (+4.5%), or one-fifth of the data for two iterative training rounds (+0.3%). Analytical experiments show instruction evolution simultaneously improves both the diversity and quality of reasoning.

Overall, our main contributions are summarized as follows:

• We propose an iterative framework to construct a CoaT-tree, and utilize rule-based rewards with backward credit assignment to form thinking-level DPO pairs for reasoning optimization.

• We introduce an instruction evolution strategy to mitigate overfitting during warm-up SFT, enhancing the model's generalization and UI understanding.

• We demonstrate the effectiveness of our method on three GUI-agent benchmarks: AITZ, AMEX, and AndroidControl. Furthermore, our approach even surpasses SoTA continual pretraining models.

## 2 RELATED WORK

### 2.1 MOBILE GUI AGENT

LLMs (Achiam et al., 2023; Xu et al., 2025a;b; Sun et al., 2024; Liu et al., 2025a) are increasingly used as autonomous agents for mobile interaction (Li et al., 2024b; Wen et al., 2023; Liu et al., 2025b; Chen et al., 2025). With the rapid development of vision-language models (VLMs), researchers build mobile GUI agents (Yang et al., 2023; Zheng et al., 2024; Qin et al., 2025; Team) and multi-agent frameworks (Ding, 2024; Li et al., 2024c; Wang et al., 2024a; Tan et al., 2025) based on closed-source VLMs. Meanwhile, some researchers focus on training agents with stronger element grounding (Cheng et al., 2024; Wu et al., 2024), page navigation (Niu et al., 2024; Lu et al., 2024; Gou et al., 2024; Wang et al., 2025), GUI understanding (You et al., 2024; Baechler et al., 2024) and task planning capabilities (Zhang et al., 2024c; Nong et al., 2024; Xu et al., 2024; Qinghong Lin et al., 2024; Dorka et al., 2024) based on open-source VLMs. Our method organizes trajectory data into multi-turns of dialogues based on the CoaT thinking pattern, preventing the agent becomes an action model with limited capabilities.

### 2.2 REINFORCEMENT LEARNING

The algorithms applied in natural language processing to align with human preferences include Direct Preference Optimization (DPO) (Rafailov et al., 2023), Identity Preference Optimization (IPO) (Azar et al., 2024), Kahneman-Tversky Optimization (KTO) (Ethayarajh et al., 2023), and

Proximal Policy Optimization (PPO) (Schulman et al., 2017). Specifically, ReFT (Luong et al., 2024) adopts reinforcement learning as a fine-tuning paradigm to improve performance on math problems. ReST-MCTS* (Zhang et al., 2024a) focuses on the higher-quality step reward, where the process reward model is important. Xie, et al. (Xie et al., 2024) label the preference via MCTS based on feedback from self-evaluation. For mobile GUI agents, Digirl (Bai et al., 2024) and Distrl (Wang et al., 2024c) use online trajectory collection to improve the generalization of agents whose process is very slow. Reachagent (Wu et al., 2025) uses DPO training to compare the quality of multiple actions. TCPO (Jiao et al., 2025) also optimizes thoughts, but does not explicitly enforce thought–action consistency. TreePO (Li et al., 2025), TreeRL (Hou et al., 2025), and SPO (Guo et al., 2025b) segment long sequences into many short segments, which leads to high computational cost and low data efficiency. In contrast, our method models thoughts with a fixed CoaT-tree and uses T-DPO to optimize the thinking process, while step values are computed directly from rule-based rewards, without unstable PRMs. This design yields more efficient sampling and training, especially in GUI-agent settings.

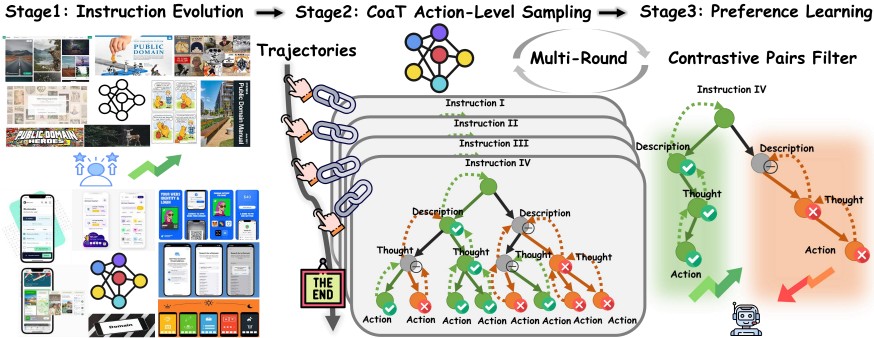

Figure 1: Overview of iterative preference learning framework. The left part presents the process of warm-up fine-tuning a general VLM to a mobile GUI domain agent with basic capabilities. The mid and right parts represent the iterative CoaT thinking-level sampling and T-DPO training process.

## 3 METHODOLOGY

In this section, we first introduce the multi-turn thinking process formulation (§ 3.1) and explain our method. As shown in Figure 1, our method starts with instruction evolution strategy (§ 3.2) to enhance output diversity in warm-up SFT stage. Then, a CoaT-tree through iterative sampling (§ 3.3) is employed for each action. Every leaf node represents a complete action and is scored using a rule-based reward function. We then backpropagate the rewards along the tree to assign credit to intermediate reasoning steps. This process yields thinking-level contrastive pairs for DPO, which further improves the model's reasoning ability. The detailed process is presented in Algorithm 1.

### 3.1 MULTI-TURN THINKING PROCESS FORMULATION

Each mobile GUI task contains a trajectory $\mathcal{T}$, several pages $u$, actions $\hat{a}$, and an instruction $I$, which can be represented as:

$$\mathcal{T} = \{I, u_0, \hat{a}_0, u_1, \hat{a}_1, \cdots, u_n, \hat{a}_n\} \tag{1}$$

We formulate action $\hat{a}_i$ in the CoaT reasoning process as a multi-turn dialogue $\hat{a}_i = [s_1, s_2, s_3, s_4]$, where $s_i$ represents description, action-thought, action-decision, and grounding, respectively. This thinking paradigm based on the thinking–decision–grounding triplet, has been widely validated as effective in previous GUI works (Shen et al., 2024; Zhang et al., 2024b; Qin et al., 2025; Cheng et al., 2024). So the reasoning process can be formulated as:

$$s_1 = \text{Description}(P_1, u_i) \tag{2}$$

$$s_2 = \text{Thought}(P_2, u_i, I, \hat{a}_0, \cdots, \hat{a}_{i-1}, s_1) \tag{3}$$

$P$ represents each round of dialogue input prompt, $I$ is the task instruction, $u$ is the current GUI, and $\hat{a}_i$ is the step $i$ history action. Agents perform poorly when decoding the entire reasoning process in a single step, which is because image modal $u$ dominates the input tokens, surpassing textual instructions $I$ and action history $\hat{a}_i$, and diverting their attention away from the textual details. During autoregressive training, the agent is unaware that producing a final answer conforming to the required format is indispensable throughout the reasoning process. Multi-turn thinking process effectively mitigates this problem, because additional dialogue steps guarantee a final answer is generated:

$$s_3 = \text{Action}(P_3, u_i, I, s_1, s_2) \tag{4}$$

$$s_4 = \text{Grounding}(P_4, u_i, I, s_1, s_2, s_3) \tag{5}$$

Previous work either performed RL in GUI-Agent directly on the trajectory without CoaT, missing the detailed thinking process of each action, or forced the model to bear the heavy burden of outputting the entire reasoning process at once. In our method, when the reasoning process ends, the final $s_4$ is recorded as $\hat{a}_{n+1}$, step $i$ moves one step forward on the trajectory $\mathcal{T}$ and its thinking step reward is calculated recursively based on final step $s_4$. Dialogue-level textual input helps balance cross-modal token proportions and steers the agent's attention toward the current reasoning step.

## 3.2 INSTRUCTION EVOLUTION

As discussed in the previous section, the CoaT patterns in the mobile agent domain are typically fixed. As a result, agents tend to overfit these static paradigms and struggle to generate diverse reasoning paths after the warm-up SFT training (as detailed in Sec 4.4). To address this issue, we enhance the original training trajectories, denoted as $\mathcal{T}$, by appending additional Q&A annotations to UI screenshots through an instruction evolution process, thereby creating a new dataset $\mathcal{Q}$ with a broader range of instruction formats. Specifically, as shown in Figure 2, the evolution process consists of three levels:

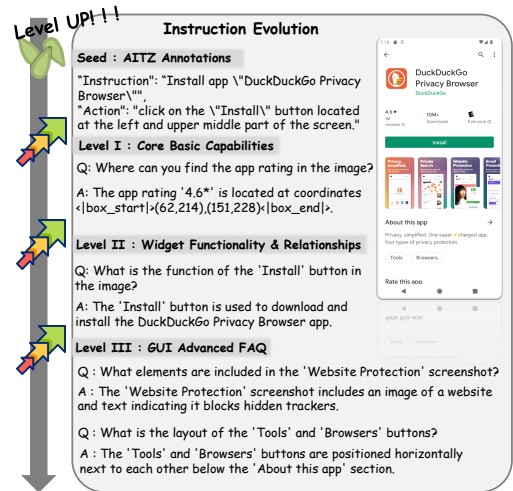

**Level I: General GUI Q&A tasks**. Grounding, Reference (Ref), and Page Descriptions are aimed at enhancing the agent's foundational capabilities. These tasks (Liu et al., 2024; Yang et al., 2024) are proven to be the core capabilities of GUI agents during the pre-training.

Figure 2: We process a three-stage instruction evolution and knowledge augmentation, enabling the agent to produce more diverse outputs for corresponding tasks while effectively mitigating overfitting.

**Level II: Widget caption and relationships**. Descriptions of widget functions and the nested partition relationships between widgets. These tasks help agents understand the relationships between widgets, as previous work (Deng et al., 2024) has found that agents tend to click on the textview, even in scenarios where the textview and the button are separate.

**Level III: GUI advanced FAQ**. Inspired by Shen et al. (2024), we design an advanced FAQ that features more complex Q&A, including descriptions of the page's structural framework as well as expectations and predictions about navigation outcomes triggered by control interactions.

**Warm-up Supervised Fine-tuning**: To develop agents with standard thinking format and expand the reasoning space, we mix $\mathcal{T}$ and the instruction evolution data $\mathcal{Q}$, then perform warm-up SFT on $\mathcal{D} = \left\{ \mathcal{T}, \mathcal{Q} \right\} = \left\{ (u, e)^{(i)} \right\}_{i=1}^{|\mathcal{D}|}$, where $u$ represents the prior knowledge (instructions, screenshot and action history) from $\mathcal{T}$ or the questions from $\mathcal{Q}$, and $e$ is the reasoning process from $\mathcal{T}$ or the answer from $\mathcal{Q}$ which is organized into multi-turn dialogues. To ensure output diversity, we select an earlier checkpoint with better potential correct space and diverse output to serve as the seed policy model. More details can be seen in Appendix B.

### 3.3 ITERATIVE PREFERENCE LEARNING

After the warm-up SFT, the agent acquires basic GUI capabilities. We construct a CoaT-tree by iteratively sampling each reasoning step and then assign a score to the leaf nodes based on a rule-based reward function. Using these scores, we generate thinking-level DPO pairs to optimize the agent's reasoning process.

**Iterative Sampling & Rule-based Reward.** We iteratively sample each reasoning step along the CoaT paradigm (Zhang et al., 2024b). The $\mathcal{K}$ sampling results $(\hat{s}_t|\hat{s}_{1:t-1})^{\mathcal{K}}$ at step $t$ can be expressed as:

$$\hat{s}_t = \left\{ \left( \hat{s}_t^{(k)} \mid \hat{s}_0, \cdots, \hat{s}_{t-1} \right) \right\}_{k=1}^{K} \tag{6}$$

Naturally, the final step in CoaT (the leaf node in the sampling tree) expresses a reward compared with the ground truth action $a^*$, which is then propagated back to other intra-nodes. The formula for the rule-based reward of leaf nodes is as follows:

$$v(s_t) = \begin{cases} 1, & s_t = a* \\ v_{type} + score_{match}, & type(s_t \sim a^*) \\ 0, & others \end{cases} \tag{7}$$

$$score_{match} = \begin{cases} v_{format} + 1 \cdot (1 - d(x,y)) - (v_{type} + v_{format}) \cdot d(x,y), & type(a*) = CLICK \\ v_{format} + (1 - v_{type} - v_{format}) \cdot F_1, & type(a*) = INPUT \\ 0, & others \end{cases} \tag{8}$$

The reward score $v(s_t)$ ranges from 0 to 1, with a fully correct prediction receiving a score of 1. We use $v_{\text{type}}$ and $v_{\text{format}}$ to indicate whether the predicted action type and output format match the ground truth. For `click` and `input` actions, we further evaluate their internal structure using smooth rewards based on spatial distance $d(x,y)$ and text match $F_1$. The final reward is computed from the similarity between the prediction and the ground truth:

- **Click:** A distance-based score between the predicted and ground-truth coordinates, normalized to $[0,1]$; smaller distances yield higher scores.
- **Input:** The $F_1$ score between the predicted and ground-truth strings; greater textual overlap yields higher scores.

The full reward is defined in Equation 7 and discussed further in Section C.

Based on the structure of the CoaT-tree, we recursively compute the value of each intermediate reasoning step. Specifically, the value of $s_{t-1}$ is computed as the average value of its $\mathcal{K}$ sampled continuations at $s_t$:

$$v(s_{t-1}) = c \cdot \frac{1}{\mathcal{K}} \sum_{k=1}^{\mathcal{K}} v(s_t^{(k)}) \tag{9}$$

Here, $\mathcal{K}$ denotes the number of sampled continuations for each reasoning step, and $c$ is a discount factor. The parameter searching experiment for $\mathcal{K}$ is described in detail in Section 4.4.

**Contrastive Data Filter.** After obtaining the sampling tree and node values, we evaluate the quality of the trees and extract contrastive data. We can divided the sampling trees into three categories $\mathcal{R} = \{\alpha, \beta, \gamma\}$ based on their output quality, and the classification standards of $\alpha, \beta, \gamma$ are as follows:

$$\alpha = \frac{\left| \left\{ \mathcal{S}^{(i)} \mid \forall v_k \in \mathcal{S}^{(i)}, v_k = 1 \right\} \right|}{\sum_{i=1}^{|\mathcal{T}|} |(u,e)^{(i)}|} \tag{10}$$

$$\beta = \frac{\left| \left\{ \mathcal{S}^{(i)} \mid \exists v_k, v_{k'} \in \mathcal{S}^{(i)}, v_k = 1, v_{k'} \neq 1 \right\} \right|}{\sum_{i=1}^{|\mathcal{T}|} |(u,e)^{(i)}|} \tag{11}$$

$$\gamma = \frac{\left| \left\{ \mathcal{S}^{(i)} \mid \forall v_k \in \mathcal{S}^{(i)}, v_k \neq 1 \right\} \right|}{\sum_{i=1}^{|\mathcal{T}|} |(u,e)^{(i)}|} \tag{12}$$

$\mathcal{S}^{(i)}$ and $v_k$ refer to the instruction $i$ sampling tree and the $k$-th leaf nodes value of $\mathcal{K}$ sampled output. $\alpha$ is considered a perfect sampling tree, which can stably output correct thoughts and actions with in-domain trajectories, $\beta$ represents potential correct trees that can be used to construct contrastive data, and $\gamma$ denotes sampling trees that require refinement. $\beta + \gamma$ is considered a valid sampling space. In $\beta$, actions with a value of 1 and as many diverse action types as possible are extracted as positive samples. In $\gamma$, the final ground truth action $a^*$ is used as a positive sample, but the intermediate steps of CoAT are not provided, and the pairs can be represented as:

$$\beta_{pairs} = \langle \hat{s}_t^{(k)} \uparrow, \hat{s}_t^{(k')} \downarrow | (\hat{s}_1, \ldots, \hat{s}_{t-1}),$$
$$v(\hat{s}_t^{(k)}) - v(\hat{s}_t^{(k')}) > 1/\mathcal{K} \rangle \tag{13}$$

$$\gamma_{pairs} = \langle a^* \uparrow, \hat{s}_t^{(k)} \downarrow | \hat{s}_1, \ldots, \hat{s}_{t-1} \rangle \tag{14}$$

**Thinking-level Direct Preference Optimization**. After CoAT thinking-level Iterative Sampling, several positive and negative example pairs are collected. During this stage, the agent policy undergoes updates through the above data-pairs, SFT loss, and CoAT-DPO loss (Rafailov et al., 2023). Suppose the agent gets values to pair $\langle +, - \rangle$ at CoAT step $t$, which are named $s_t^+$ and $s_t^-$; we have the agent performing a comparison for these pairs based on the same thoughts $s_{1:t-1}$, which can be calculated as:

$$\mathcal{L}_{\text{T-DPO}} = -\mathbb{E}_{(s_{1:t-1}, s_t^-, s_t^+) \sim \mathcal{T}_s} \left[ \log \sigma(\beta \log \frac{\pi_\theta(s_t^+|s_{1:t-1})}{\pi_{ref}(s_t^+|s_{1:t-1})} \right.$$
$$\left. -\beta \log \frac{\pi_\theta(s_t^-|s_{1:t-1})}{\pi_{ref}(s_t^-|s_{1:t-1})}) \right], \tag{15}$$

To further refine the agent's performance post-optimization, we employ the updated agent as the new base agent to continue collecting contrastive CoAT-action level pairs for additional T-DPO training. This iterative process is maintained until the agent reaches the performance bottleneck.

---

**Algorithm 1:** Iterative CoAT thinking-level sampling and DPO self-training.

---

**Input:** base VLM $\pi$, advanced annotated model $R_{SoTA}$, step-level trajectory data $\mathcal{T}$, instruction evolution Q&A set $\mathcal{Q}$, number of sampling $\mathcal{K}$, golden action $a^*$, value function $v$, the sampled CoAT data $D$, number of iterations $\mathcal{N}$.

1: **for** $i = 1$ to $N_0$ **do**
2:     $\mathcal{Q}^* \leftarrow$ instruction_evolution($R_{\text{SoTA}}, \mathcal{T}$) // instruction evolution by GPT-4o
3:     $\mathcal{Q} \leftarrow$ human_evaluation($h, \mathcal{Q}^*$) // human filter
4: **end for**
5: $\pi_{S_0} \leftarrow$ Warm-up_SFT($\pi, \mathcal{T}, \mathcal{Q}$) // fine-tune seed model
6: **for** $n = 1$ to $\mathcal{N}$ **do**
7:     **for** $i = 1$ to $|\mathcal{T}|$ **do**
8:        $D_i \leftarrow$ generate_sampling_thought($\pi_{S_{n-1}}, \mathcal{T}_i, \mathcal{K}$) // CoAT Sampling
9:        $V_i^{leaf} \leftarrow v(D_i, a_i^*)$ // match and calculate leaf values using Eq(7)
10:       $V_i^{intra} \leftarrow$ recursive_calculate($D_i, V_i^{leaf}$) // recursive intra node values using Eq(9)
11:       $D_i^+, D_i^- \leftarrow$ contrastive_data_filter($D_i, V_i$) // filter positive and negative data using Eq(13, 14)
12:     **end for**
13:     $\pi_{S_n} \leftarrow$ DPO($\pi_{S_{k-1}}, D^+, D^-$) // DPO self-training reference model
14: **end for**
**Output:** $\pi_S, D_G, \mathcal{Q}$

---

# 4 EXPERIMENTS

## 4.1 EXPERIMENTS SETUPS

**Dataset. AITZ** (Zhang et al., 2024b) is a high-quality trajectory set filtered and re-annotated from AITW (Rawles et al., 2023), containing four subsets , which also includes five types of actions **AMEX** (Chai et al., 2024) uses the same apps and action space as AITZ, but its task instructions are more complex and detailed, with an average trajectory length of 15+. **AndroidControl** (Li et al., 2024a) includes OOD datasets, such as app unseen and task unseen.

Table 1: **Main results of AITZ dataset.** ZS, FT, PF, and IPL are short for zero-shot, fine-tuning, specific domain pre-training, and iterative preference learning, respectively. '-' represents that the agent or evaluation prompt is not open-sourced. Seed means the seed model for sampling and T-DPO training. $R_i$ refers to the number of iterations during training.

| Model | Mode | Atomic | | | | | | | | |
| | | SCROLL | CLICK | | TYPE | | PRESS | STOP | Total | |
| | | | type | match | type | match | | | type | match |
| --- | --- | --- | --- | --- | --- | --- | --- | --- | --- | --- |
| CogAgent (CoaT) | ZS | 70.22 | 88.23 | 66.15 | 45.80 | 21.80 | 45.95 | 24.60 | 72.59 | 53.28 |
| AUTO-GUI (CoaT) | FT | 61.40 | 74.56 | 32.20 | 87.20 | 81.40 | 57.70 | 74.40 | 82.98 | 47.69 |
| AriaUI-MoE | FT | 53.73 | 85.51 | 60.20 | 84.20 | 80.80 | 63.70 | 76.38 | 78.53 | 63.56 |
| Seeclick-7B | PF | 11.14 | 69.92 | 52.96 | 53.80 | 53.00 | 67.88 | 55.36 | 62.93 | 49.11 |
| UGround-7B | PF | 58.22 | 80.94 | 58.48 | 82.56 | 73.85 | 58.22 | 68.78 | 74.54 | 60.19 |
| OS-Atlas-7B | PF | 76.12 | 75.82 | 54.83 | 87.80 | 81.60 | 68.67 | **81.75** | 77.83 | 65.11 |
| UI-Tars-7B | PF | 52.50 | 83.03 | 64.27 | **89.97** | 82.76 | 61.87 | 74.35 | 77.59 | 65.61 |
| Falcon-UI-7B | PF | - | - | - | - | - | - | - | **84.70** | 69.10 |
| Qwen2-VL-7B (CoaT) | FT | 47.50 | 81.53 | 59.72 | 81.96 | 73.85 | **58.22** | 67.39 | 74.26 | 60.36 |
| AITZ-Seed | FT | 42.83 | 82.48 | 53.16 | 82.56 | 75.29 | 56.65 | 61.82 | 73.14 | 55.40 |
| MobileIPL | IPL | 51.08 | **91.73** | **71.45** | 88.20 | **83.40** | 51.69 | 78.17 | 81.90 | **69.15** |

Table 2: **Main results on AMEX**. Seed means the seed model for sampling.

| Model | Training Data | Gmail | Booking | Music | SHEIN | News | CM | ToDo | Signal | Yelp | Overall |
| --- | --- | --- | --- | --- | --- | --- | --- | --- | --- | --- | --- |
| SeeClick-7B | AITW+External | 28.2 | 29.4 | 18.1 | 20.0 | 30.0 | 53.1 | 30.7 | 37.1 | 27.4 | 30.44 |
| SphAgent-7B | AITW | 32.1 | 45.9 | 46.1 | 35.1 | 48.3 | 61.1 | 55.9 | 43.3 | 42.9 | 45.63 |
| SphAgent-7B | AMEX | 61.7 | 68.2 | 77.7 | **72.0** | 71.9 | 64.6 | **79.6** | 71.3 | 69.6 | 70.71 |
| AriaUI-MoE | AMEX | 63.1 | 62.3 | 68.5 | 58.9 | 83.0 | 54.7 | 62.5 | 83.3 | 66.9 | 64.10 |
| UGround-7B | AMEX | 70.9 | 68.8 | 72.7 | 63.7 | 77.7 | 67.7 | 63.7 | 80.1 | 67.6 | 69.12 |
| SphAgent-7B | AITW + AMEX | 62.4 | 68.1 | 76.3 | 71.9 | 68.6 | 67.3 | 77.6 | 66.0 | 64.1 | 69.14 |
| OS-Atlas-7B | AMEX | 61.1 | **73.5** | 77.9 | 61.6 | 75.2 | 66.4 | 71.0 | 75.9 | 72.0 | 70.33 |
| UI-Tars-7B | AMEX | 67.7 | 70.0 | 71.8 | 63.8 | 71.5 | 67.7 | 77.0 | **86.4** | 72.8 | 70.33 |
| Qwen2-VL-7B | AMEX | 58.0 | 70.1 | 76.6 | 63.8 | 79.4 | 66.8 | 67.8 | 80.2 | **76.6** | 69.01 |
| | + CoaT | 75.9 | 68.1 | 77.7 | 66.2 | 76.8 | 66.4 | 77.5 | 79.6 | 65.6 | 70.93 |
| MobileIPL-7B | AMEX (Seed) | 57.0 | 60.2 | 68.8 | 63.1 | 75.0 | 50.2 | 65.6 | 77.7 | 62.6 | 62.19 |
| | MobileIPL | **77.3** | 71.8 | **80.0** | 68.4 | **85.3** | **71.3** | 73.5 | 82.1 | 71.8 | **74.29** |

**Metrics.** For evaluation, we use **Step.Acc** as metrics, consistent with Auto-GUI(Zhang & Zhang, 2023), measures the agent's performance and uses **Action Type** to assess the degree of action type matching. This metric effectively evaluates the model's planning ability.

**Baselines.** Following prior work(Wu et al., 2024)(Qin et al., 2025), we use Qwen2-VL-7B (Wang et al., 2024b) as the backbone of our model. We select CogAgent (Hong et al., 2024), AUTO-GUI, Shpagent, OS-Atlas, UGround, UI-Tars and FedMobileAgent as baseline agents. GUI continuous pre-training agents can be further divided into two categories: (1) training the model as a GUI grounding agent, such as OS-Atlas-7B. (2) training the model as a general GUI agent, such as UI-Tars. More details are provided in Appendix D.

## 4.2 MAIN RESULT

**AITZ.** As shown in Table 1, MobileIPL achieves SoTA performance on most metrics. The reason for the lower PRESS Acc. is discussed in Section 4.4 and Appendix H. Multiple rounds of T-DPO improve MobileIPL by more than 10% (55.40% -> 69.15%) compared to the seed model MobileIPL and Qwen2-VL-7B (60.36% -> 69.15%). Compared to continuous pre-training agents such as Falcon-UI, which is pre-trained on three million GUIs, MobileIPL still surpasses a performance difference of 0.05%. The amount of training data required by our method is substantially smaller than that used by these pre-training approaches.

**AMEX.** As shown in Table 2, MobileIPL surpasses the previous SOTA model, SphAgent-7B, by 3.58%. It also outperforms the baseline model (Qwen2-vl+CoaT) by 3.36%. Additionally, MobileIPL surpasses OS-Atlas (+3.69%) and UI-Tars (+3.69%), both of which also use Qwen2-vl as the backbone. With the incorporation of CoaT, the baseline model Qwen2-vl shows an increase of 1.92%,

demonstrating the effectiveness of CoaT patterns. In summary, these results confirm that MobileIPL delivers significant improvements over existing models in long trajectory scenarios.

Table 3: High-level instruction experiment results on **AndroidControl**.

| Mode | Model | Grounding | Step.Acc |
|------|-------|-----------|----------|
| FT | Aria-UI-7B | 43.2 | 10.2 |
| | InternVL-2-4B | 72.7 | 66.7 |
| | Qwen2-VL-7B (SFT) | 68.5 | 69.1 |
| PF | OS-Atlas-7B | 78.5 | 71.2 |
| | Falcon-UI-7B | - | 72.7 |
| | UI-Tars-7B | **80.5** | 72.5 |
| RL | Qwen2-VL-7B(GRPO) | 70.7 | 69.8 |
| Ours | MobileIPL | 77.0 | **72.7** |

Table 4: High-level instruction results on **AndroidControl in-domain** and **OOD** subsets.

| Mode | Model | IDD | app-UN | task-UN |
|------|-------|-----|--------|---------|
| FT | PaLM 2S(full) | 65.5 | 58.7 | 59.7 |
| | PaLM 2S(LoRA) | 70.8 | 58.5 | 59.6 |
| | Qwen2-VL-7B(SFT) | 69.1 | 61.4 | 64.1 |
| PF | FedMobileAgent | 54.7 | 52.3 | 51.2 |
| | SphAgent-7B | 69.4 | 57.1 | 62.9 |
| | OS-Atlas-7B | 71.2 | 60.7 | 66.2 |
| RL | Qwen2-VL-7B(GRPO) | 70.2 | 68.1 | 69.7 |
| IPL | MobileIPL-7B | **73.6** | **70.0** | **72.2** |

**AndroidControl.** As shown in Table 3, MobileIPL achieves SOTA performance in Step.Acc (72.7%), reaching the SOTA model Falcon-UI with fewer data. MobileIPL also outperforms continual pre-training agents in the GUI domain, such as OS-Atlas (+1.5%) and UI-Tars (+0.2%). Compared to the baseline model Qwen2-VL(SFT), MobileIPL not only improves Mobile Agent performance but also enhances grounding by 8.5%. As shown in Table 4, MobileIPL continues to achieve SOTA performance in unseen OOD settings, demonstrating strong generalization. In contrast, compared to performance in the IDD domain, the pre-trained model OS-Atlas shows a significant drop. MobileIPL exhibits less performance degradation in out-of-domain settings. We also ran GRPO with Qwen2-VL under the same computational resources, and found OOD performance similar to MobileIPL, because both are self-training. However, MobileIPL still outperforms GRPO in all subsets.

## 4.3 ABLATION STUDY

To test the effectiveness of IPL and instruction evolution, we conducted ablation experiments. First, removing IPL and using only SFT caused performance to drop from 65.4% to 60.4%, compared to the first round of MobileIPL, highlighting the crucial role that IPL plays. Next, removing instruction evolution led to a 2.5% drop in IPL performance in the first round. This occurs because, without evolution, the model generates fewer training samples (156,418 -> 113,239). And as shown in Figure 3 (a), without instruction evolution, the diversity of model outputs decreased, causing a drop in IPL performance. This further confirms that instruction evolution is crucial for improving IPL.

Table 5: Ablation study results on AITZ.

| Model | Scroll | Click | Type | Press | Total |
|-------|--------|-------|------|-------|-------|
| MobileIPL-R1 | 45.8 | 71.1 | 81.2 | 23.5 | 65.4 |
| - IPL | 46.9 | 59.4 | 78.6 | 55.4 | 60.4 (-5.0) |
| - Evo (R1) | 44.8 | 67.7 | 78.8 | 24.0 | 62.9 (-2.5) |
| - IPL Negative (R1) | 46.9 | 61.1 | 74.2 | 56.6 | 61.4 (-4.0) |
| - IPL + Naive DPO (R1) | 47.5 | 59.7 | 73.8 | 58.2 | 60.3 (-5.1) |
| - 1/2 training data (R1) | 42.9 | 68.3 | 79.0 | 43.8 | 64.8 (-0.6) |
| - 4/5 training data (R2) | 30.8 | 67.1 | 77.6 | 33.2 | 60.6 (-4.8) |

**Ablation Study.** Additionally, we remove negative samples from IPL-R1, training the model using only fully correct samples. This results in a 4.0% performance drop, suggesting that negative samples help the model learn how to reason rather than merely memorize (SFT). Furthermore, training on the entire trajectory with naive DPO reduces performance from 65.4% to 60.3%. Compared with SFT trained on CoaT tree positive data (–IPL Negative), naive DPO is still 1.1 % lower, confirming the effectiveness of CoaT-tree sampling and the thinking-process optimization.

**Low Resource.** We also perform low-resource on AITZ, sampling 1/2 and 1/5 of the training data. As shown in table 5, using only half of the data, the first round of IPL training already outperformed

the best results achieved by the original CoAT-SFT (-IPL) and naive DPO training. Furthermore, when using only one-fifth of the data, the second round of IPL training surpassed the performance of CoAT-SFT (-IPL), demonstrating the effectiveness of our method even in low-resource scenarios.

## 4.4 Discussion and Analysis

**Rollout Efficiency and Performance Trade-off.** We compare MobileIPL with GRPO and the MCTS-style baseline SPO-Chain(Guo et al., 2025b). Although MobileIPL requires more rollout sampling than GRPO, it achieves better accuracy. Compared with SPO-Chain, MobileIPL uses only about half as many rollouts per sentence ($\sim$27 vs. $\sim$54) while still obtaining a +1.12 accuracy gain. We also observe that SPO-Chain is sensitive to hyperparameters (e.g., temperature and cut-point probabilities). Its best performance is achieved with a higher temperature (1.4) to enlarge the exploration space, but this also produces longer sentences and slows down sampling. Overall, MobileIPL offers a better efficiency–performance trade-off and is more practical for GUI settings.

Table 6: Comparison of RL methods in terms of accuracy and rollouts per sentence.

| Model | Accuracy | Rollouts per sentence |
|---|---|---|
| MobileIPL (Ours) | 69.15 | $\sim$27 |
| SPO-Chain | 68.03 | $\sim$54 |
| GRPO | 66.29 | 8 |

**Reasoning Space Sampling.** To evaluate the instruction evolution, we analyze the diversity of the sampling space for **Random 1000 steps**, the standard deviation of encoded embeddings, the dimensionality-reduced distribution, and the distribution of $S^{(i)}$ mentioned in Section 3.3. As shown in Figure 3 (a), the thoughts after instruction evolution exhibit a broader space than direct SFT. Additionally, the embedding standard deviation within each tree increases significantly compared to the original data (+ 0.158). The diversified outputs do not negatively impact the agent's reasoning process, while the proportion of action sampling that includes the correct answer improves from 72.7% to 77.9%. The bottom-right subplot reflects the distribution of output accuracy. **Consistently Correct** indicates that all samples for the current step match the golden answer, while **Consistently Error** is the opposite. **Both** represents cases where some samples are correct while others are incorrect, which serves as an ideal source for constructing T-DPO pairs. Compared to 47% on the evolved data, the agent achieves 68.7% convergence on the original data but exhibits a strong polarization(4%). Three-stage instruction evolution significantly expands the sampling space (from 4% to 31%), proving that it simultaneously improves both the diversity and quality of reasoning. More details are in Appendix F.

**Parameters Searching.** We conducted an ablation study on the impact of the sampling number ($\mathcal{K}$) per stage and iterative round number ($R$). As shown in the table 7, increasing the number of samples generally leads to better model performance. However, since our framework adopts a tree structure, increasing the sampling number from 3 to 4 causes the minimum number of tree nodes to grow significantly from $3^3 = 27$ to $4^3 = 64$. Despite this sharp increase, the performance improvement is limited (less than 1%). Therefore, we adopt a sampling number of 3 for the final

Table 7: **Parameters Searching** on AITZ for the first round. $\mathcal{K}$ is the sampling number and R is the round of T-DPO learning.

| Parameter | TYPE | CLICK | SCROLL | PRESS | STOP | Total |
|---|---|---|---|---|---|---|
| $\mathcal{K} = 2$ | 79.2 | 68.9 | 35.1 | 38.3 | 76.1 | 64.0 |
| $\mathcal{K} = 3$ | 81.2 | 71.1 | 45.8 | 23.4 | 73.5 | 65.3 (**+1.3**) |
| $\mathcal{K} = 4$ | 81.2 | 70.4 | 51.2 | 35.5 | 66.8 | 65.9 (**+0.6**) |
| $R = 0$ | 75.7 | 53.3 | 43.7 | 58.2 | 63.2 | 57.5 |
| $R = 1$ | 77.5 | 71.1 | 43.3 | 23.5 | 67.0 | 61.2 (**+3.7**) |
| $R = 2$ | 80.5 | 71.1 | 47.0 | 31.1 | 67.6 | 64.1 (**+2.9**) |
| $R = 3$ | 82.0 | 71.5 | 47.2 | 47.8 | 79.1 | 68.4 (**+4.3**) |
| $R = 4$ | 82.6 | 71.5 | 51.1 | 51.7 | 78.2 | 69.2 (**+0.8**) |

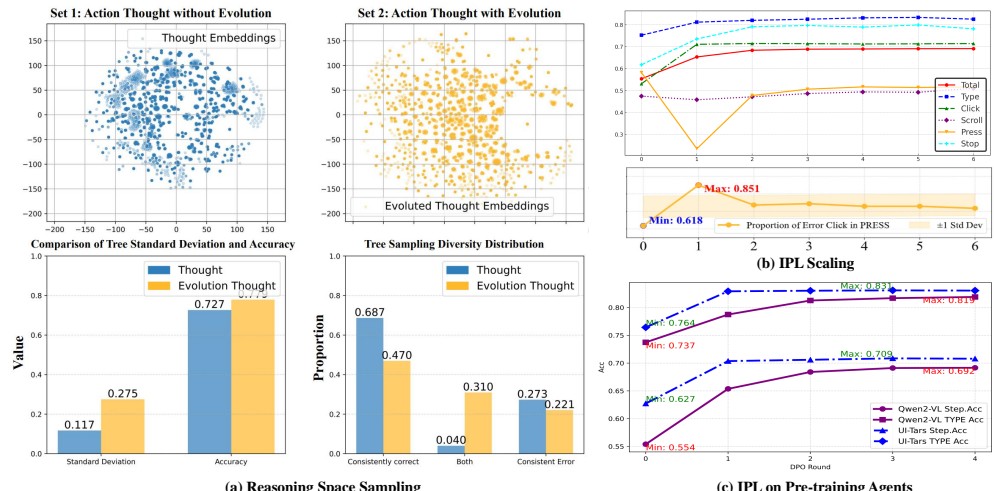

Figure 3: (a) Reasoning diversity before and after instruction evolution (left top) and the distribution of standard deviation and accuracy (left bottom). (b) **Step.Acc** changes for action types in AITZ across IPL iterations (right top). (c) The performance of UI-Tars-7B and Qwen2-VL-7B on AITZ as the seed model with 4-round IPL training (right bottom).

experiments. Regarding the number of rounds, we observe that both IPL performance and the size of the self-training dataset converge after several iterations. We therefore select the convergence round as our default setting. Additional details on computational cost are provided in Appendix G.

**IPL Scaling.** Although overall Step.Acc increases across IPL iterations, not all action types follow this trend. As shown in Figure 3(b), from the seed model to the first IPL round, PRESS accuracy drops sharply (58.22% → 23.49%), whereas CLICK rises (53.26% → 71.12%). In the second round, however, PRESS accuracy rebounds. This stems from the severe underrepresentation of PRESS actions early on: the proportion of PRESS samples in the preference data grows from 1.6% (1 round) to 10.9% (2 round) as training progresses. With greater reasoning diversity and more PRESS-related examples, the model gradually learns PRESS behaviors and recovers accuracy in later rounds.

**Iterative Preference Learning On GUI Continuous Pre-training Agent.** As discussed in the previous experimental analysis, continuous pre-training in the GUI domain provides the agents with a stronger base model. However, we still need to explore the compatibility between post-training IPL, instruction evolution, and pre-training. As shown in Figure 3 (c), UI-Tars outperforms Qwen2-VL-7B in all training stages, demonstrating better performance during the instruction evolution phase (62.7% > 55.4%). After four rounds of IPL, UI-Tars Step.Acc improves by 1.4% compared to MobileIPL (69.2% -> 70.6%). More importantly, UI-Tars nearly converges after the first round of IPL, significantly reducing the number of sampling and preference learning iterations, thereby keeping the computational cost of post-training within an acceptable range.

## 5 CONCLUSION

In this paper, we propose Mobile Iterative Preference Learning (**MobileIPL**), a self-training GUI agent framework that incorporates instruction evolution, iterative sampling in the CoaT-tree, and a rule-based reward. We extensively evaluate MobileIPL on the AITZ, AMEX, and AndroidControl benchmarks, demonstrating its effectiveness. Furthermore, MobileIPL exhibits strong generalization capabilities on the OOD subsets of AndroidControl. Experiments show that instruction evolution increases output diversity, generates more training data in IPL, and thereby improves IPL performance. Finally, Continuous Pre-training experiments confirm the mutual reinforcement between MobileIPL and pre-training, leading to enhanced performance.

## 6    ETHICS STATEMENT

We have rigorously refined our dataset to remove any elements that could compromise personal privacy, thereby guaranteeing the highest level of protection for individual data. Instruction evolution was completed by AI SoTA close-sourced VLM, to whom we paid the necessary compensation to ensure that the training data was not leaked. The human evaluation of our work was carried out through a meticulously randomized selection of IT professionals. This process ensured a gender-balanced and educationally diverse panel, reflecting a wide spectrum of perspectives and expertise.

## 7    REPRODUCIBILITY STATEMENT

All models and datasets used in this paper are open-source. The full experimental setup is detailed in Appendix D. Unless noted, all experiments use the same settings. We describe compute resources in Appendix G. Overall, these practices make our results reproducible.

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

## A  COAT THINKING PROCESS

The table 8 summarizes the CoT paradigms (inputs and outputs) used in prior related works. CoaT paradigm for fine-tuning agents: The effectiveness of the approach in AITZ stems from the inclusion of extra screen descriptions as part of the input, along with joint output of Screen Context, Action Think, Action Target, and Action Result. In contrast, our experiments show that a stage-wise multi-turn dialogue output leads to better performance. In this setup, the model focuses on a single sub-task at each stage, which not only improves clarity but also encourages a simplified and deliberate reasoning process. This insight aligns with UI-TARS, which only requires the model to generate a brief thought during inference. Small-scale agent framing: Even models with relatively small parameter sizes can benefit from task-decomposed downstream training. For instance, OS-ATLAS and Falcon-UI adopt a similar architecture using GPT-4o for textual description and OS-ATLAS-base as the grounding model. They fine-tune models separately on different downstream tasks, resulting in a collection of OS-ATLAS-pro models, each specialized for a specific sub-task. Large-scale prompting-based frameworks: Larger models typically adopt a multi-agent framework to support a CoT-style reasoning process. For example, AppAgent v2 and Mobile-Agent-v2 both utilize a

plan–action–reflection architecture to complete tasks. In our work, we adopt a stage-wise CoaT multi-turn dialogue format, where the model focuses on one sub-task at a time. This design enables us to move away from the dependence on extra screen description inputs, as seen in AITZ, while leveraging the description + grounding structure proposed in OS-ATLAS to form the final structure of the MobileIPL CoaT paradigm.

Table 8: Comparison of methods and their corresponding CoT paradigms.

| Method | Backbone | Input | Output (CoT Paradigm) |
|---|---|---|---|
| Android in the Zoo | One fine-tuned agent | Instruction + image + screen description | Screen Context → Action Think → Action Target → Action Result |
| UI-TARS | One fine-tuned agent | Instruction + image | Thought → Action |
| OS-ATLAS | One fine-tuned agent + one large agent | Instruction + image | Action Description + Action Model |
| Mobile-Agent-v2 | Multi-agent + prompt engineering | Instruction + image | Stage 1: Plan → Stage 2: Action → Stage 3: Reflection |
| MobileIPL | One fine-tuned agent | Instruction + image | Stage 1: Description → Stage 2: Plan → Stage 3: Action → Stage 4: Grounding |

## B  SELECTION OF SEED POLICY MODEL

In our preliminary experimental exploration, we discovered that for the seed policy model, better performance in the SFT phase does not necessarily translate to a higher upper bound in the subsequent IPL phase. This is because as training progresses, the model's output space becomes increasingly aligned with the training data, reducing its diversity in sampling. Consequently, for incorrect instances, the model tends to generate erroneous outputs regardless of the sampling attempts. To address this, we propose a sampling-oriented selection method for the seed policy model, incorporating the following two evaluation metrics:

**Sampling Accuracy**($Acc_S$), which requires the model to hit more correct actions $a$ in the sampled output space $\mathcal{S}$.

$$Acc_S = \frac{\sum_{i=1}^{|\mathcal{T}|} \left| \left\{ e_j^{(i)} \mid a^{(i)} \sim e_j^{(i)}, e_j^{(i)} \in \mathcal{S}^{(i)} \right\} \right|}{\sum_{i=1}^{|\mathcal{T}|} |\mathcal{S}^{(i)}|} \tag{16}$$

**Sampling Diversity**($Div_R$), which requires the model to have a more diverse and extensive sampling space. Standard deviation calculation of a single sampled tree $\mathcal{D}ev_{S^{(i)}}$:

$$Dev_{S^{(i)}} = \frac{1}{T} \sum_{t=1}^{T} \text{StdDev} \left( \mathbf{E}(\hat{s}_t^{(k)}) \mid k = 1, \ldots, \mathcal{K} \right) \tag{17}$$

Among them, $\mathbf{E}(\hat{s}_t^{(k)})$ represents the representation of the $k$th sample output of the $t$th step after the encoder. Calculation of the standard deviation of the set $Div_S$:

$$Div_R = \frac{1}{N} \sum_{i=1}^{N} Dev_{S^{(i)}} \tag{18}$$

where $N$ is the number of sampled trees in the set $\mathcal{R}$.

## C  RULE-BASED REWARD DESIGN

**Derivation Of The Value Function.** Our value function incorporates hyperparameters inspired by ReFT and is also influenced by the sampling number used during IPL. We explain the rationale behind key parameter choices in our method, especially those in Eq.(7), Eq.(13), and Eq.(14):

**Strong Reward**: We follow the ReFT (Luong et al., 2024) score to define strong reward signals, assigning values of 1 and 0, corresponding to fully correct and completely incorrect reasoning paths, respectively. In ReFT, a supervision signal of 0.1 encourages the model to produce a final answer following the predefined format. In our approach, this signal is repurposed to reward action type

Table 9: Text F1 vs. text-embedding similarity: ablation on reward design. Best results are in **bold**.

| Reward Type | TYPE (type) | TYPE (acc) | TOTAL (type) | TOTAL (acc) |
|:---:|:---:|:---:|:---:|:---:|
| BERT | 84.17 | 76.70 | 77.81 | 62.49 |
| F1 | **87.78** | **81.23** | **78.74** | **65.37** |

matching. Meanwhile, an additional $v_{format}$ reward is introduced to encourage proper formatting of actions.

**Weak Reward**: For input action, the value linearly increases from $v_{format} + v_{type}$ up to the strong reward level, with $v_{format}$ acting as a threshold to distinguish weak from type-correct reward. For grounding actions, values range between $v_{format} + v_{type}$ and 1, too. A value of $v_{format}$ indicates minimal correctness (e.g., extractable coordinates), while 1 indicates a closest match with the golden action, suitable for DPO pairing. Except $v_{format}$ and $v_{type}$ serving as discrete supervision signals, all other value signals are maintained as continuous. $1 / \mathcal{K}$ in Eq. (13) arises naturally from our hierarchical training structure. For example, if one child is incorrect (e.g., value drops from 1 to 0), the average value for the parent node decreases by 0.33 when the sampling number is 3. Thus, $1 / \mathcal{K}$ serves as a meaningful threshold to distinguish positive vs. negative examples in the CoAT tree.

**Text F1 vs. Text-Embedding Similarity:** We replaced the F1-based reward with a BERT-based semantic reward and evaluated both variants. As shown in Table 9, the F1 reward outperforms the BERT embedding reward across all metrics, with the largest gain on *TYPE ACTION (acc)* (+4.53%). This aligns with the importance of exact keyword matching in GUI input, indicating that F1 is better suited than semantic similarity for reward design in mobile UI input scenarios.

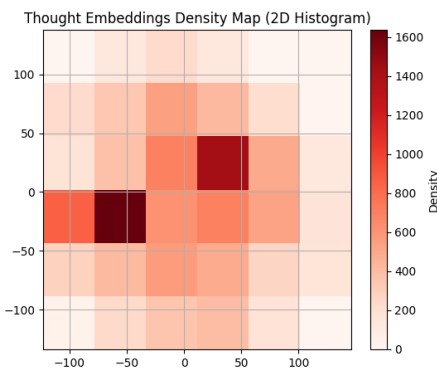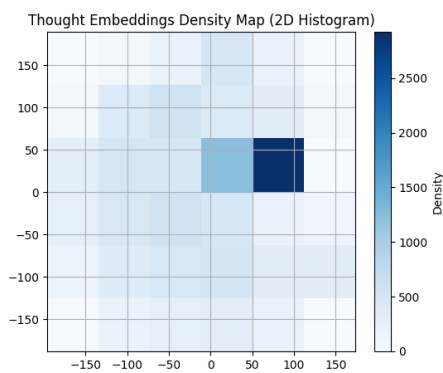

Figure 4: The heatmap at the left represents the sampling before instruction evolution, while the one at the right represents the sampling after instruction evolution.

## D    EXPERIMENT SETUP

**Models.** Unlike AITZ, we do not compare the CoaT result with the expected page and decide whether to roll back because most actions in real-device scenarios cannot be rolled back without cost. Previous work conducted continual pretraining on Qwen2-VL-7B using GUI domain data, resulting in a stronger base model. In our ablation study, we discuss the impact of continuous pretraining on IPL. ()

**Setup.** We conduct hyperparameter searches on AITZ to reproduce the baseline results and find that the optimal learning rate ranges from 3e-5 to 3e-6. Therefore, all baseline fine-tuning experiments adopt this setting. Before IPL, during the instruction evolution stage, we apply LoRA fine-tuning with a LoRA rank of 128. For IPL Stage 1, we use a learning rate between 1e-6 and 1e-7. In subsequent stages, we apply a constant learning rate of 1e-7. The batch size is consistently set to 128. During

fine-tuning (including baseline fine-tuning), we enable ViT training, whereas in the IPL phase, we experiment with freezing ViT. For AITZ training, we followed the Falcons' approach, utilizing a maximum 1540×1540 resolution. For other experiments, we reduce the resolution to 1280×720 to optimize computational efficiency. The maximum context length is set to 32K for all experiments. The fine-tuning experiments are conducted for 2 epochs, while IPL training is performed for 1 epoch. Since the large volume of Android control data, we sample 1/5 of the dataset for each IPL training iteration.

**CoaT Multi-turn Dialogue Prompts.**

1. **Page Description.** *Based on the mobile screenshot: Image URL, identify and describe the key elements visible on the screen, including any text, buttons, icons, input fields, or other interactive components.*

2. **CoaT Action Thought.** *Given the task: instruction, and considering the contextual details from the image alongside the full history of previous actions: action history, determine the most logical and effective next step. Focus on providing a clear, actionable, and goal-oriented response to advance the task.*

3. **CoaT Action Description.** *Task: Determine the Most Appropriate Next Step. Based on the previous analysis and the objective, determine the most appropriate next step to achieve the goal. Choose from the following options: - **click**: Select a button or specific UI element by specifying it clearly (e.g., 'click xxx', where 'xxx' is the button name or identifier). - **scroll**: Perform a scrolling action if the required element is not visible, specifying the direction (e.g., 'scroll up', 'scroll down'). - **type**: Input specific text into a field or search bar, specifying the text clearly (e.g., type "content"). - **press**: Interact with device-level buttons such as Home, Back, or Enter, specifying the button (e.g., "press Back"). - **stop**: Conclude the task, indicating that the objective has been achieved. Provide the chosen action in the specified format and ensure it aligns with the analysis and the visible UI elements.*

4. **Click Action Grounding.** *As discussed earlier, your task now is to identify the precise screen region coordinates to tap for the action coat action. The coordinates must be integers and strictly within the range of 0 to 1000 for both axes. Please provide your response in the required format: <\box_start\>(top_x, top_y),(bottom_x, bottom_y)<\box_end\>. Ensure your output adheres to these constraints and remains concise.*

**Instruction evolution Prompts.**

1. **Page Description Annotation.** *I will provide you with a mobile page. Please describe the current page. Your description should include the content of the page and its general functionality. Please note that the descriptions you generate should be of moderate length. Your page description should match the actual image.*

2. **Action Thought Annotation.** ***QUERY**: task, **ACTION HISTORY**: To proceed with the query, your past actions include: action history, **NEXT ACTION**: This is the next action you need to take: coat action, **TASK**: Given the screen and the above information, you have three tasks to do. First, you have to analyze what you have done. Second, you should analyze the screen for relevant details that might pertain to the given query. This includes checking for specific applications, icons, or buttons that are visible and any information or results that are currently displayed on the screen. Tip: If the screen does not have the information you need, you can scroll left or scroll up to try to get the information. Don't answer this logic question by saying that because the provided **NEXT ACTION** is..., therefore, the next action is... You need to think carefully on your own. You must answer the question with suitable lengths and the following format: 'Think: I have done..., Current screen is..., I need to... So the next action is ...' Your final action should be the same as the NEXT ACTION above.*

3. **Q&A Annotation.** *Your goal is to draw inspiration from the given images and image description information to create multiple new questions and answers. This new creation is closely related to the given image and information, but the answers involved should be directly derived from the given information, because UI positions and UI text are one-to-one correspondence. Specifically, you should construct the following three types of questions*

*and answers, a total of 15: 1. the function of some elements in the image. 2. Grounding questions and answers (the coordinates and approximate location of the target in the image). 3. Partial detailed information questions and answers (the structural relationship between multiple elements, type, style, etc.). Please try to keep your questions and answers diverse and informative, and ignore the message in the device status bar. Here is the information related to the image: UI positions: {ui positions}, UI text: {ui text}, coat screen desc: {coat screen desc}, Please provide the following information in JSON format with the key questions and answers, and Don't add annotation parsing:*

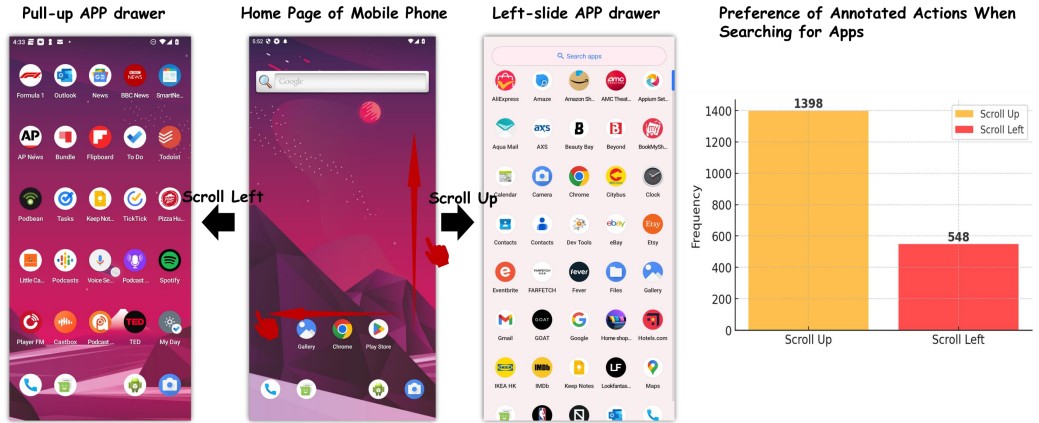

Figure 5: The left figure shows an example of unstable annotation preferences in **AMEX**, while the right figure presents the proportion of this type of annotation.

## E  HUMAN ANNOTATION

**SoTA Model Cost**: We use GPT-4o for annotation, which is priced at 4 USD per 1M input tokens and 16 USD per 1M output tokens. As shown in Appendix B (Instruction Evolution Prompts), each image has a resolution of 1080 × 2440. The input prompt is approximately 2K tokens, and the output is around 0.5K tokens, resulting in a per-page annotation cost of 0.008 USD (input) + 0.004 USD (output) = 0.012 USD. Given that the Android in the Zoo dataset [1] contains around 18,000 pages, the total annotation cost is approximately 200 USD.

**Human Verification**: Human verification is used to identify cases where the model produces incorrect thoughts due to being forced to align with the golden action. As shown in following table, some initial annotations contain template-like phrases (e.g., "Since", "annotation") and mention multiple possible actions. These mistakes often happen when the model tries to justify a given action even if it doesn't match the actual screen.

The bolded text in following Table highlights common errors. To address this, we regenerated the prompts and applied automatic filtering based on specific tokens, which helps reduce the need for manual work.

**Before Human Verification:** "coat_action_think": "Reason for the Action: - To find and compare prices, the user needs to locate similar products. - Swiping up or scrolling down is a logical step to explore further information or discover additional options, as apps like Etsy often display "similar items" or "recommended products" below the primary product listing. - This action aligns with typical app design patterns where related items or comparison options are placed further down the page. 4. Logical Derivation of Action: - Since the task specifies comparing prices of similar items, and the current screen does not explicitly list such items, the user must navigate downward to uncover more options. - The instruction to "swipe up" (or scroll down) matches the action type ("action_type": "scroll", "direction""down"), indicating an attempt to reveal additional content or sections. 5. **Since the next action swiping down matches the annotation, the next action is swiping down**.

**After Human Verification:** Think: I have previously navigated "Back," which indicates that I may have returned to the previous screen or listing, likely as part of exploring options related to the query

Table 10: Data size and training time across iterative rounds.

|  | Standard SFT | Round 1 | Round 2 | Round 3 | Round 4 | Round 5 |
|---|---|---|---|---|---|---|
| Data Samples | 63,158 | 156,418 | 34,010 | 28,491 | 17,027 | 17,780 |
| Time (hours) | 2.8 | 7.9 | 2.0 | 1.7 | 1.0 | 1.0 |

for comparing prices of men's athletic shoes. Current screen is displaying detailed information about a product on the Etsy app. This includes item details such as "Handmade item," "Made to order," and shipping information. There are also options to "Follow shop," "Message seller," and "Add to cart," but it does not appear to directly show other comparable items or prices for similar men's athletic shoes.I need to scroll further down on this screen to potentially uncover additional details, such as related products, a comparison section, or other useful information for the query. So the Next action is Scroll down."

## F    ITERATIVE SAMPLING IN THE COAT-TREE

As shown in Figure 4, before instruction evolution, the distribution is highly concentrated, with only 8 points exceeding 1000 (including 3 points above 1200). After instruction evolution, the distribution becomes more balanced, with 20 points exceeding 1000 (including 2 points above 1500).

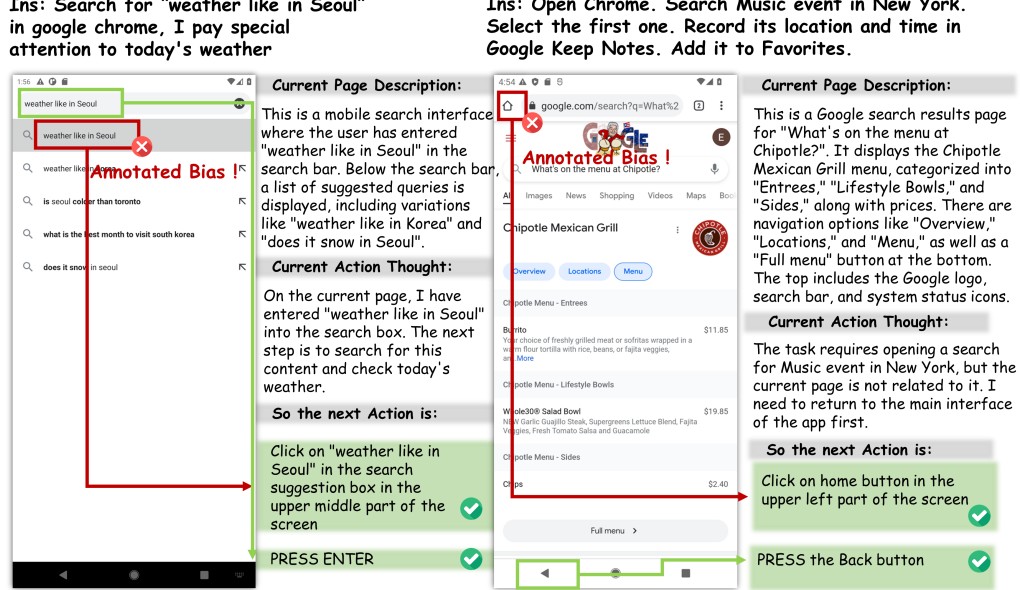

Figure 6: An example from **AITZ** demonstrates that when the task and image are the same, multiple actions may navigate to the same page.

**Potential correct space ratio.** The proportion of $|\alpha| + |\beta|$ represents the potential correct space on the training data, and the change of this metric can clearly express the agent's ability to repair and reason out the correct process based on the correct answer.

## G    COMPUTATIONAL COST OF IPL ACROSS ITERATIONS

As shown in Table 10, while the first IPL iteration incurs higher cost due to the larger volume of preference data, subsequent iterations are significantly lighter. The training time per round decreases rapidly, as the model generates fewer low-quality samples and requires fewer updates. In fact, by the third iteration, the training time becomes comparable to or even lower than the initial supervised fine-tuning (SFT) stage.

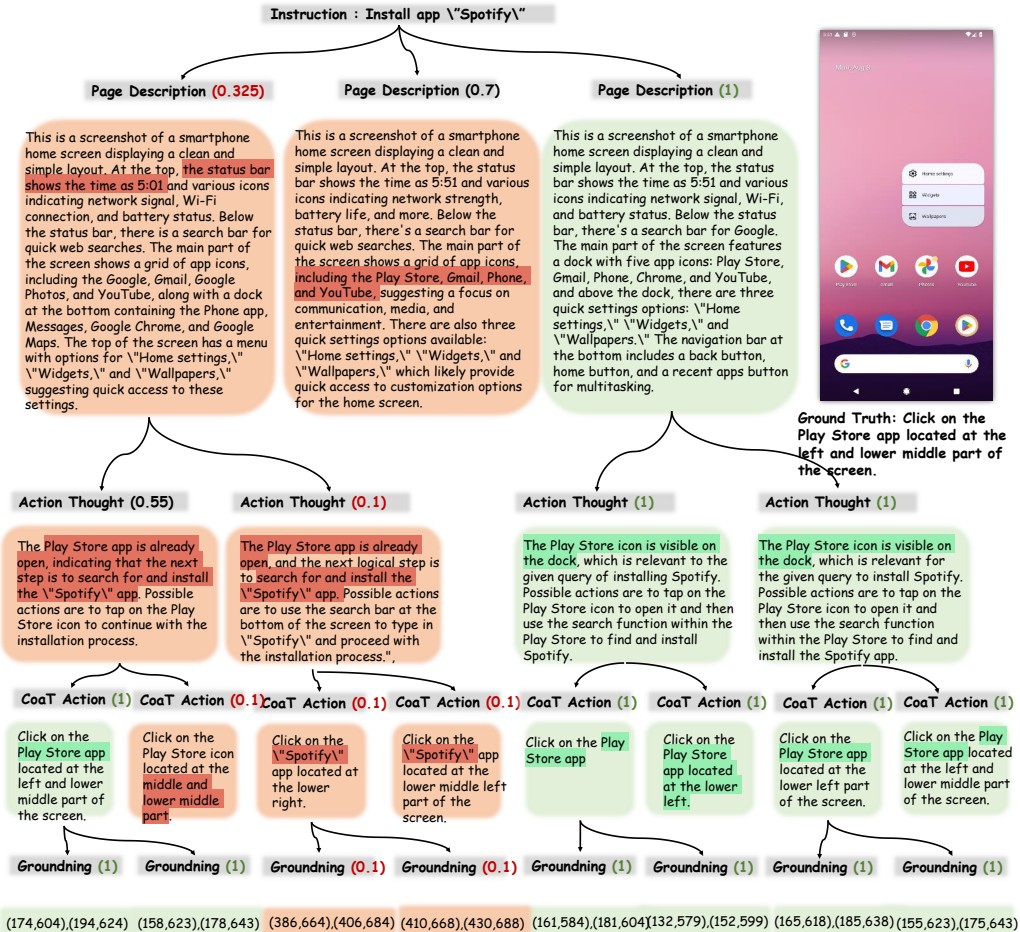

Figure 7: A sampling tree from **AITZ** demonstrates how the value is calculated.

Therefore, the overall compute overhead of IPL remains moderate and manageable, especially considering its performance gains. Compared to SFT, IPL introduces only a modest increase in compute, but brings substantial improvements in reasoning and generalization.

## H CASE STUDY

**Unstable annotation preferences.** As shown in Figure 5, the left section illustrates two different annotation preferences when searching for an app from the Home Page: **SCROLL UP** and **SCROLL LEFT**, leading to different destination pages. The right part shows the overall preference distribution when annotators need to find an app. In rare cases, the annotation involves clicking on Google Play Store to perform a search. This phenomenon is quite common because, fundamentally, the task completion paths for a UI Agent are diverse. This is also the key difference between online evaluation and offline data evaluation. From this, we observe that RL training on data with unstable preferences performs worse than SFT (e.g., AITZ SCROLL). This is because the DPO pair training method inherently attempts to correct errors in sampled preferences. As a result, the agent oscillates between two decisions when encountering the same GUI and instruction, failing to achieve consistent alignment.

**Action Equivalence.** Unlike Unstable Annotation Preferences, where different actions lead to different but equivalent pages, the issue here arises from annotators' random labeling habits in the training data, preventing the model from learning a consistent preference. Action Equivalence refers to the phenomenon where multiple actions on the same page can lead to the target page. However, since only one action is annotated as correct, other valid actions are mistakenly treated as incorrect. As

shown in Figure 6, after entering a search query, clicking on a suggested item in the recommendation bar, and pressing the Enter key on the keyboard produce the same effect. Similarly, when navigating back, clicking the on-screen back button and pressing the hardware back button yield the same outcome.

**Thinking-Level Sampling.** As shown in Figure 7, unlike mathematical reasoning, the CoaT process may not exhibit clear logical or computational errors. For a given action, a sampling CoaT data may produce hallucinations (Page Description) due to insufficient detail in the page description or fabricated elements; generate repetitive thoughts (Action Thought) due to neglecting action history; describe the wrong relative position of the correct element (CoaT Action); or misgrounding an element (Grounding), which is then classified as a negative sample. At the same time, outputs with more detailed and accurate descriptions, diversified thoughts, and different ways of describing the same widget are classified as positive samples. Negative examples may be disadvantageous compared to positive examples, for example, because the description of the page is not detailed enough or the positioning of the elements is not accurate enough. At the same time, the wrong process may also give the correct result, but this is a very rare case. In this example, negative samples are generated due to the following three reasons: (1) **Rough page description:** The page contains eight app icons, but the agent's description includes only four apps: Play Store, Gmail, Phone, and YouTube; (2) **Hallucinated Thought:** The agent is unclear about its current page location. In reality, it is on the Home page, but it mistakenly believes it is in the Play Store (e.g., "The Play Store app is already open"). (3) **Fabricated Position and Elements:** The agent generates the action "Click on the 'Spotify' app", even though there is no Spotify icon on the current page. This hallucination may stem from the instruction. Additionally, the Play Store icon should be located at the lower left part of the screen, but the agent incorrectly describes it as being in the middle and lower middle part.

# I   USAGE OF LLM STATEMENT

This paper utilized an LLM to improve the clarity and fluency of the text.

