# OpenReview forum: "MobileIPL: Enhancing Mobile Agents Thinking Process via Iterative Preference Learning"
_ICLR.cc/2026/Conference — ICLR 2026 Poster_

### Official Review · Reviewer_Lxqd · 2025-10-28

**Soundness:** 3
**Presentation:** 3
**Contribution:** 3
**Rating:** 6
**Confidence:** 3

**Summary:**

This paper introduces MobileIPL, a novel framework to enhance the reasoning abilities of VLM-based mobile agents for tasks involving mobile GUIs. Addressing limitations in existing approaches, such as overfitting from supervised fine-tuning and reliance on costly manual step-level reward annotation, the authors propose Iterative Preference Learning, which constructs diverse reasoning trajectories through Monte Carlo Tree Search and uses rule-based scoring for intermediate thoughts instead of process reward models. Feedback from scored leaf nodes is backpropagated to earlier reasoning steps, forming thinking-level preference pairs for Direct Preference Optimization to improve both final action selection and the reasoning process itself. To boost generalization and prevent overfitting, a three-stage instruction evolution strategy generates varied Q&A pairs grounded in real UI screenshots, enhancing agents’ contextual understanding.

**Strengths:**

- The paper introduces a creative method (MobileIPL) for improving reasoning in mobile GUI agents, leveraging an iterative sampling and learning process that avoids expensive and hard-to-scale process-level manual annotation. By combining Monte Carlo Tree Search (MCTS) with rule-based rewards and backward value propagation, the paper sidesteps unstable process reward models and manual step annotation, streamlining the learning process.
- Unlike many prior works that optimize only for final task reward/accuracy, this method explicitly scores and optimizes at the level of intermediate "thinking" steps, fostering better structured and generalizable agent reasoning.
- The approach is thoroughly evaluated on three widely recognized benchmarks, using both in-domain and out-of-domain tasks, and consistently outperforms several strong recent baselines and continual pretraining models. The analysis demonstrates the benefits of each introduced component (e.g., iterative rounds, instruction evolution), strengthening the validity of the methodological choices.

**Weaknesses:**

- The method entirely relies on rules (format, type, spatial distance, and F1 for input) to quantify step-level reward. These may not capture nuanced or partially correct reasoning paths, edge-case actions, or subtle mistakes that a learned process reward model or human judgements might identify. Is the rule-based approach robust to complex/ambiguous UI layouts or fuzzy matching of instructions? Even with more granular step-wise signals, rule-based reward systems are susceptible to reward hacking (the model may exploit rules to maximize reward without genuinely improving reasoning).
- The iterative sampling and tree construction (MCTS-like CoaT-tree) can be computationally expensive, scaling poorly with both the number of sampled trajectories (K) and the depth of reasoning. Can the authors report the runtime and resource requirements, especially compared to simple SFT or even RL-based pipelines?
- The process involves choices of sampling number K, discount factor c, DPO data filtering thresholds, etc. There is insufficient discussion of robustness to these parameters, or general guidance about their selection for new domains.
- Although the instruction evolution is partly filtered by humans, it remains heavily reliant on LLM-generated QA pairs. Are these synthetic instructions always realistic and helpful? Does the method generalize if downstream tasks differ greatly from the synthetic questions generated?

**Questions:**

See Weaknesses.

---

> ### Author Response · Authors · 2025-11-23
> **Response to Reviewer Lxqd**
>
> # Response to Reviewer Lxqd
>
> We thank the reviewer for the detailed and thoughtful comments. Below we respond to the main weaknesses.
>
> > W1: Robustness of rule-based reward under complex / ambiguous UI layouts
>
> A1: Our main goal is to fix the GRPO-style issue of "only optimizing actions but not thoughts". Unlike GRPO, where thought and action are generated and trained together, MobileIPL first generates a fixed-structure CoaT thought and then conditions the action on this thought. In this setup, a bad thought is much more likely to produce bad actions than a good thought.
>
> Even though the reward itself is rule-based, we sample multiple continuations for each thought and use backward value propagation on the CoaT-tree to evaluate whether a thought is good or bad, and then apply DPO to move the model toward better thoughts. In practice, this combination (multi-sample evaluation + tree backprop + DPO) makes the rule-based scheme sufficiently robust, even when the UI layout is complex or slightly ambiguous.
>
>
> > W2: Computational cost of iterative sampling and CoaT-tree construction
>
> A2: MobileIPL uses a fixed multi-turn dialogue structure, which allows us to reuse KV cache across different branches of the tree, significantly reducing the effective generation cost.
>
> Consider a simple example where we need to roll out 4 CoaT sequences:
>
> - text1 → text2 → text3
>
> - text1 → text2 → text3'
>
> - text1 → text2' → text3''
>
> - text1 → text2' → text3'''
>
> With KV cache reuse, we only need to newly generate:
>
> - text1 once,
>
> - text2 and text2' once each,
>
> - text3, text3', text3'', text3''' once each.
>
> So the total number of forward generations is about 1 + 2 + 4 = 7, instead of 4 × 3 = 12 if we were to generate each full sequence independently as in a naive GRPO-style rollout.
>
> Compared to SFT, IPL indeed incurs additional sampling cost. However, given the substantial performance gains, we find this overhead acceptable. Compared to GRPO-style RL, the fixed CoaT structure + KV reuse makes MobileIPL’s sampling process competitive in practice rather than prohibitively expensive.
>
> > W3: Sensitivity to K, discount factor c, and data filtering thresholds
>
> A3: In the paper, we discuss the choice of sampling number K in table 6, and find that larger k generally helps but with diminishing returns, so we finally set K=3. Here we additionally report a small hyperparameter study for the discount factor c and the data filtering threshold (margin in DPO pairing), under K=3:
>
> | c   | Data threshold | Match (Acc) |  Training pairs |
> |-----|----------------|-------------|------------------|
> | 1.0 | 1 / K          | 65.4        | 156,418          |
> | 0.9 | 1 / K          | 63.5        | 130,571          |
> | 0.8 | 1 / K          | 62.4        | 123,315          |
> | 1.0 | 2 / K          | 64.3        | 127,290          |
> | 1.0 | 1 / (2K)       | 61.1        | 167,367          |
>
> We observe:
>
> - As the discount factor c decreases, both the effective data size and performance drop.
>
> - Raising the threshold above 1/K removes too many useful pairs and hurts performance, while lowering it below 1/K adds many low-contrast pairs that act as noise and likewise degrade performance.
>
> Based on this, we choose c=1.0 and threshold = 1/K as a good trade-off. While some sensitivity exists, the method is reasonably stable around these settings, and the trends provide practical guidance for new domains.
>
>
> > W4: Generalization of instruction evolution when downstream tasks differ from synthetic Q&A
>
> A4: To study this, we conducted an additional experiment on AITZ click data by converting it into a "user instruction → coordinate"QA format, e.g.:
>
>     > "Given 'I want to go to Chrome', which point should be clicked"
>
> We compare SFT with and without instruction evolution:
>
> | Setting          | ACC   | pass@5 |
> |------------------|-------|--------|
> | SFT only         | 72.98 | 74.08  |
> | + Instruction Evo| 73.56 (+0.58) | 75.50 (+1.42)|
>
> We find that:
>
> - The improvement in greedy ACC is relatively small.
>
> - However, pass@5 shows a clearer gain, indicating that instruction evolution enlarges the potential correct space and prevents the model from collapsing to a narrow SFT solution.
>
> This suggests that even when the synthetic Q&A distribution differs from the exact downstream tasks, the evolved instructions still help by diversifying the reasoning space and improving the model’s potential, which is particularly beneficial for the subsequent IPL stage.

---

> ### Author Response · Authors · 2025-11-27
> **Response to Reviewer Lxqd**
>
> Dear Reviewer Lxqd,
>
> Thank you again for taking the time to review our paper. As the rebuttal period is approaching its end, we wanted to kindly check whether our responses have addressed your concerns.
>
> If there is anything that is still unclear or any additional feedback you would like us to address, we would be very happy to provide further clarification.
>
> Best regards,
>
> The paper 23626's Authors

---

### Official Review · Reviewer_6kud · 2025-10-31

**Soundness:** 2
**Presentation:** 2
**Contribution:** 2
**Rating:** 4
**Confidence:** 3

**Summary:**

#### Main Content Summary

* **Problem Statement**: VLM-based mobile agents rely on the **Chain of Action-Planning Thoughts (CoaT)** paradigm, but the **scarcity of diverse CoaT trajectories** severely restricts their generalization ability. Existing self-training solutions either overlook the correctness of **intermediate reasoning steps** or require **expensive process-level annotations** to construct Process Reward Models (PRMs).
* **Proposed Solution (MobileIPL)**: The paper introduces **MobileIPL (Iterative Preference Learning)**. This method is designed to enhance the agent's thinking process by **iteratively learning preferences** to distinguish between high-quality and low-quality reasoning paths. This mechanism effectively optimizes the correctness of intermediate thoughts **without the high cost of manual process-level labeling**.
* **Observed Failure Modes**: Analysis of negative samples reveals consistent reasoning errors, including **Rough page description**, **Hallucinated Thought** (e.g., believing the agent is on the Play Store when it is on the Home page), and **Fabricated Position and Elements** (e.g., generating an action for a non-existent element).

**Strengths:**

#### Strengths (1-2 points)

* **1. Cost-Effective and Scalable Process Improvement**
    * **Detail**: MobileIPL successfully addresses the bottleneck of requiring **expensive process-level annotations** for building Process Reward Models (PRMs). By utilizing an Iterative Preference Learning approach, the method provides a scalable and cost-efficient mechanism to automatically guide and improve the quality of the agent's intermediate reasoning steps.

* **2. Direct Focus on Intermediate Reasoning Quality**
    * **Detail**: Unlike many self-training strategies that only optimize for the final task success rate, MobileIPL's core mechanism explicitly targets the **correctness, detail, and diversity of the "thoughts" (intermediate reasoning steps)**. This helps the agent build a more robust and trustworthy foundation for planning and execution.

**Weaknesses:**

#### Weaknesses (3-4 points)

* **1. Failure to Address Root Causes of Hallucination**
    * **Detail**: The paper highlights severe errors like **Hallucinated Thought** and **Fabricated Elements**. While MobileIPL attempts to suppress these errors via preference learning, it **does not fundamentally solve** the VLM's underlying limitations in **visual grounding** and accurate internal state tracking, leaving the model susceptible to imagining non-existent states or elements.

* **2. High Domain Specificity and Limited Generality**
    * **Detail**: The proposed method is highly specialized for **VLM-based mobile agents** operating within the structured environment of **GUI tasks**. The direct applicability of MobileIPL's preference learning strategy may be restricted outside of this specific domain, such as in continuous control, physical robotics, or less-structured web navigation.

* **3. Risk of Error Accumulation in Iterative Training**
    * **Detail**: As an iterative self-training technique, the approach carries an inherent risk of **compounding errors** or training instability. If the preference model in earlier iterations inaccurately favors flawed reasoning patterns, these errors could be amplified throughout the process, leading the final agent to converge on a suboptimal or unreliable policy.

* **4. Constraint by the CoaT Architectural Paradigm**
    * **Detail**: The entire framework is built upon enhancing the existing **Chain of Action-Planning Thoughts (CoaT)** structure. This reliance prevents the model from exploring or learning potentially more efficient **non-sequential, parallel, or highly flexible** reasoning processes that might be required for solving complex, multi-faceted tasks more optimally than a strict linear chain allows.

**Questions:**

### Open Research Questions for MobileIPL

* **1. Generalization Beyond GUI and Abstract Action Spaces**
    * **Question**: How can the Iterative Preference Learning (IPL) framework be successfully extended and validated in **more complex, less-structured embodied environments**, such such as physical robotics or 3D navigation, where actions are continuous and visual states are highly dynamic and less deterministic than in a GUI?

* **2. Mitigating the Root Cause of Hallucination in the Thought Process**
    * **Question**: Can MobileIPL's approach be integrated with **explicit visual grounding mechanisms** or **internal state tracking models** to not just penalize (via preference learning) but fundamentally **prevent** the generation of Hallucinated Thoughts and Fabricated Elements?

* **3. Optimizing the Preference Learning Signal for True Causality**
    * **Question**: How can the preference signal be refined to better distinguish between a **causally correct** but less detailed thought sequence and a **highly detailed** but ultimately incorrect or inefficient one, especially in cases where a flawed process yields a correct final result (the rare case mentioned in the paper)?

---

> ### Author Response · Authors · 2025-11-23
> **Response to Reviewer 6kud**
>
> # Response to Reviewer 6kud
>
> > Q1: Generalization Beyond GUI and Abstract Action Spaces
>
> A1: Although our experiments focus on GUI agents, note that the action space in GUI control is already partially continuous. For example, a CLICK action is parameterized by a 2D coordinate on the screen, which lies in a continuous spatial domain.
>
> In MobileIPL, we explicitly design a smooth reward for such continuous parameters: the click reward is based on the distance between predicted and ground-truth coordinates, rather than a binary 0/1 signal. Our experiments show that this design leads to a clear improvement for click actions:
>
> | Model                 | Click (match) |
> |-----------------------|---------------|
> | Qwen2-VL (baseline)   | 59.72         |
> | MobileIPL (ours)      | 71.45         |
>
>
> > Q2: Mitigating the Root Cause of Hallucination in the Thought Process
>
> A2: Standard RL for GUI agents typically supervises only the final outcome/action and does not directly optimize the thoughts, which encourages hallucinated reasoning. MobileIPL, built on the CoaT paradigm, explicitly constructs thought-level DPO pairs via sampling and value backpropagation, so hallucinated or inconsistent thoughts are directly penalized during training. While this does not change the base VLM architecture, it substantially reduces hallucinations induced by the training objective itself.
>
> > Q3: Optimizing the Preference Learning Signal for True Causality
>
> A3: In our view, both less detailed thoughts that ignore key elements and incorrect thoughts that deny their existence are bad for causal planning, even if they occasionally yield the correct action. MobileIPL performs multi-sample reasoning with MCTS-style credit assignment, and thoughts that repeatedly lead to lower-value outcomes are treated as negative in DPO pairing. Thus, IPL explicitly pushes the model away from such unreliable thoughts and toward causally informative thoughts that consistently support correct actions.

---

> ### Author Response · Authors · 2025-11-27
> **Response to Reviewer 6kud**
>
> Dear Reviewer 6kud,
>
> Thank you again for taking the time to review our paper. As the rebuttal period is approaching its end, we wanted to kindly check whether our responses have addressed your concerns.
>
> If there is anything that is still unclear or any additional feedback you would like us to address, we would be very happy to provide further clarification.
>
> Best regards,
>
> The paper 23626's Authors

---

### Official Review · Reviewer_5WvC · 2025-11-01

**Soundness:** 2
**Presentation:** 2
**Contribution:** 2
**Rating:** 4
**Confidence:** 3

**Summary:**

The paper proposes MobileIPL, a self-training framework for vision-language mobile GUI agents that enhances reasoning via Iterative Preference Learning. It introduces a CoaT-tree built through Monte Carlo sampling, uses rule-based rewards with backward credit assignment to form thinking-level DPO pairs, and mitigates overfitting via a three-stage instruction evolution strategy. Experiments on AITZ, AMEX, and AndroidControl show strong performance and out-of-domain generalization.

**Strengths:**

- The CoaT-tree construction via iterative sampling enables fine-grained reasoning optimization without manual step annotations and seems novel.

- Experimental results on AITZ, AMEX, and AndroidControl indicates that MobileIPL outperforms previous strong baselines with less data usage.

**Weaknesses:**

- A few recent related works are not well-discussed. For example, TCPO [1] proposes thought-centric preference optimization, which is similar to the Thinking-level DPO proposed in this work. TreePO [2], TreeRL [3], SPO [4] all introduces tree-structure rollout and value backpropagation, which is similar to the iterative sampling process of MobileIPL.

-  The clarity is not clear. For example, (1) Section 3.3 is not a complete part. It introduces Iterative Preference Learning. However, this section ends with the construction of $\\beta _ \\mathrm{pairs}$ and $\\gamma _ \\mathrm{pairs}$, without the preference optimization process. Although the DPO process is mentioned in Algorithm 1, it is not clear enough for clarity. Also, it is not clearly demonstrated what $\\alpha$, $\\beta$, $\\gamma$ mean, are they sampling trees (Line 260) or numbers (Line 248) or qualities (Line 247)? (2) what is $d(x,y)$ in Line 223? It is not explained before or after. **Make sure all the notations or concepts are clearly explained before using them.**

- The motivation behind Iterative Preference Learning is unclear. Specifically, if an outcome-based reward is accessible and the values of intermediate reasoning steps can be obtained via MCTS, why not directly apply GRPO or PPO with step-level or segment-level advantage estimation—as in SPO [4]? Such an approach would provide finer-grained reward signals than DPO and is also simpler. Could the authors please provide further justification or experimental results to clarify this design choice?

[1] TCPO: Thought-Centric Preference Optimization for Effective Embodied Decision-making. EMNLP 2025

[2] TreePO: Bridging the Gap of Policy Optimization and Efficacy and Inference Efficiency with Heuristic Tree-based Modeling

[3] TreeRL: LLM Reinforcement Learning with On-Policy Tree Search. ACL 2025

[4] Segment Policy Optimization: Effective Segment-Level Credit Assignment in RL for Large Language Models. Neurips 2025

**Questions:**

Please see the weaknesses for the major concerns.

---

> ### Author Response · Authors · 2025-11-23
> **Response to Reviewer 5WvC**
>
> # Response to Reviewer 5WvC
> We thank the reviewer for the constructive feedback and the recognition of our CoaT-tree design and empirical improvements. Below we address the raised concerns point by point.
>
> > W1: Relation to TCPO, TreePO, TreeRL, and SPO
>
> 1. Difference from TCPO:
> - TCPO also samples trajectories to compare "good" and "bad" thoughts, but under single-sample generation per thought, it can still produce mismatched pairs such as (t^-, a^+), where the thought is suboptimal but the final action happens to be correct.
> - Our CoaT-tree explicitly organizes multi-turn CoaT reasoning for each GUI action and performs tree-structured sampling over this space, enabling more consistent and structurally grounded thinking-level comparisons in the GUI setting.
>
> 2. Difference from TreePO, TreeRL, and SPO.
>
> They are primarily developed for **text-only** tasks, and their segmentation strategy is defined at arbitrary positions. This design implies that a single sentence often needs to be split into many segments and re-sampled multiple times.
>
> In our reproduction of SPO-Chain on AITZ, we observe that:
> - Each sentence is split into about 6 segments on average.
> - With a rollout budget of 9 per segment, this leads to roughly 6×9=54 generations per sentence.
>
> By contrast, MobileIPL uses a fixed CoAT structure to define the stages of Description → Thought → Action, which:
> - Requires only 3^3 = 27 node expansions per sentence in our CoaT-tree.
> - Treats the thought as prior knowledge when generating the final action, effectively decoupling and then jointly optimizing thought and action, which helps avoid thought–action inconsistency.
>
> > W2: Clarity and notation
>
> We appreciate  pointing out the clarity issues in Section 3.3.
>
> Due to space limitations, we initially only presented the DPO step in Algorithm 1 and did not describe the preference optimization process in detail in the main text. We have now added explicit textual descriptions of the DPO-based optimization following the construction of positive/negative pairs, and clarified how it operates on the CoaT-tree in the revised paper.
>
> We believe these revisions substantially improve the readability and remove the ambiguity.
>
> > W3: Why not GRPO/PPO with step-level or segment-level advantages as in SPO?
>
> We do not directly apply GRPO/PPO with step-level or segment-level advantages (as in SPO). Our reasons are as follows:
> 1. Efficiency considerations:
>
> Using GRPO/PPO directly on an MCTS-style rollout with segment-level credit assignment is computationally expensive in our GUI setting. For SPO-Chain on AITZ, we empirically observe:
>
> - Each sentence is split into ~6 segments.
>
> - Each segment requires a full rollout of 9 .
>
> - This leads to about **54** rollouts per sentence.
>
> In contrast, MobileIPL’s CoaT structure requires only about **27** expansions per sentence, and the fixed structure allows KV-cache reuse, providing better sampling efficiency for GUI tasks.
>
> 2. Thought–action consistency.
> - If we apply GRPO/PPO naively, the model typically generates thought and action jointly in one long sequence, so the optimizer directly acts on the combined output. This makes it difficult to explicitly enforce consistency between the reasoning and the eventual action.
> - In MobileIPL, we explicitly treat the thought as a prior: the thought is generated in previous question, then fed back into the model as part of the context for action generation. This staged design encourages the final action to be consistent with the previous thought and allows us to optimize thought and action in a more structured way.
> 3. Data utilization and supervision quality.
>
> In SPO-style RL, each gradient update typically uses only the segments from a **single** sentence, and segments with all-zero or all-one rewards are hard to distinguish in terms of learning signal. Moreover, if the first sampled sentence does not contain a good trajectory, that update provides almost no useful direction for policy improvement.
>
> In contrast, our DPO-based IPL:
> - Naturally filters out trees where all leaf values are 0 or all 1.
> - Focuses on contrastive pairs derived from CoaT-tree nodes with genuinely different values, which improves the informativeness of each update.
> - Constructs DPO pairs from **all valid nodes in the tree**, so each rollout contributes richer supervision, helping the model converge faster and more stably.
>
> Empirically on AITZ:
> |Method|Step.Acc|Rollouts per sentence|
> |-|-|-|
> |Ours| 69.15| 27|
> |SPO-Chain|68.03|~54|
> |GRPO|66.29| 8|
>
> We find that SPO-Chain requires about 54 rollouts per sentence, while MobileIPL needs only 27, yet still achieves +1.12 higher acc. SPO-Chain is also sensitive to hyperparameters (e.g., temperature, cut-point probabilities): its best performance appears when increasing the temperature to 1.4 to enlarge the exploration space, which makes sentences longer and slows sampling. Overall, IPL provides a better efficiency–performance trade-off and is more practical in GUI setting.

---

> ### Author Response · Authors · 2025-11-27
> **Response to Reviewer 5WvC**
>
> Dear Reviewer 5WvC,
>
> Thank you again for taking the time to review our paper. As the rebuttal period is approaching its end, we wanted to kindly check whether our responses have addressed your concerns.
>
> If there is anything that is still unclear or any additional feedback you would like us to address, we would be very happy to provide further clarification.
>
> Best regards,
>
> The paper 23626's Authors

---

### Official Review · Reviewer_YpsF · 2025-11-03

**Soundness:** 3
**Presentation:** 3
**Contribution:** 2
**Rating:** 6
**Confidence:** 3

**Summary:**

This work introduces MobileIPL, a self-training framework for mobile GUI agents that enhances the intermediate “thinking” or reasoning steps of vision-language models interacting with mobile interfaces. The key innovations are (1) instruction evolution — using a model like GPT‑4o to generate diverse Q&A pairs from real mobile UI screenshots to enrich training data and avoid reasoning collapse; (2) construction of a CoaT-tree (Chain of Action-Planning Thoughts) via iterative sampling of reasoning trajectories at the action level, scoring leaf nodes through rule-based rewards, and back-propagating values to generate thinking-level preference pairs (T-DPO) for training; and (3) iterative rounds of preference learning to improve the agent’s reasoning diversity and correctness without requiring heavy process-level annotations. Empirical results on GUI benchmarks (AITZ, AMEX, AndroidControl) show that MobileIPL outperforms strong baselines (like continual-pretraining agents) and generalizes better to out-of-domain mobile UI tasks.

**Strengths:**

The GUI agents are becoming more and more popular recently, which is a very interesting and highly practical direction.

The overall presentation and paper writing is very clear. The pipeline components, i.e., SFT data collection and preference training annotation and filtering, are all illustrated well, which are helpful in advancing open-source GUI agents development.

The results are comprehensive and convincing, especially demonstrating OOD performance.

**Weaknesses:**

From a novelty standpoint, the primary contributions of this work lie in building a comprehensive end-to-end pipeline, whereas most of the technical components appear to be adaptations of existing methods or relatively straightforward extensions, especially in light of the recent surge of research on agentic system design and RL-based training. I sincerely appreciate the considerable engineering effort invested in large-scale SFT data collection and RL system implementation—this is clearly valuable for the open-source community and not trivial to execute. Nevertheless, the limited methodological innovation remains the main reason I am unable to assign a higher score.

**Questions:**

My primary concern, as noted in the weaknesses section, relates to the level of technical innovation in the proposed approach. I would welcome further clarification from the authors regarding the novel methodological contributions and how they advance beyond existing agent-based and RL-training frameworks.

---

> ### Author Response · Authors · 2025-11-23
> **Response to Reviewer YpsF**
>
> # Response to Reviewer YpsF
>
> Thank you very much for your positive evaluation of our work. Below we further clarify our contributions and compare them with existing agent-based and RL-training frameworks.
>
> > W1: Clarification regarding the novel methodological contributions
>
> 1. **Thought-level supervision via CoaT-Tree.**
>    MobileIPL constructs a CoaT-Tree to evaluate intermediate thoughts and generate explicit thought-level supervision. The dialog-based formulation enforces consistency between thoughts and subsequent actions.
>
> 2. **Smooth reward for parameterized actions.**
>    For parameterized actions such as *click* and *type*, we introduce a smooth, distance-aware reward function that captures partial correctness rather than using binary outcomes, providing more stable and informative gradients.
>
> 3. **Three-stage instruction evolution.**
>    We propose a three-stage instruction evolution pipeline that progressively refines trajectories, thoughts, and actions, yielding higher-quality supervision and improving model robustness.
>
> Overall, MobileIPL enables more structured and fine-grained reasoning supervision than existing GUI-agent frameworks.
>
>
> > W2: How we advance beyond existing agent-based and RL-training frameworks
> 1. Compared with traditional RL (e.g., GUI-R1 / GRPO-style training):
> - These methods typically judge reward solely based on the final action, ignoring the quality of the intermediate thoughts. This can lead to reward hacking, where the model optimizes only the action output while neglecting the reasoning process, which is harmful for performance. In addition, the reward is usually binary (0/1), and does not fully exploit the rich structure of actions such as CLICK and TYPE that contain parameters.
>
> - We construct a CoaT-Tree to obtain **think-level supervision** for optimizing the model’s intermediate thoughts,  and design a **smooth, structured reward** based on the distance for CLICK actions and the similarity for TYPE actions, providing more fine-grained supervision and better shaping of the learning signal.
>
> 2. Compared with agent frameworks (e.g., Mobile-Agent-v2 [2]):
> - Frameworks such as Mobile-Agent-v2 can be very powerful, but they often rely on closed-source large models and tool ecosystems, which limits reproducibility and deployment in purely open-source settings.
>
> 3. Empirical comparison on AITZ:
> - As shown below, our method outperforms pure GRPO-style RL, while agent-framework approaches need much larger models to reach similar performance:
>
> | Model                               | Backbone / Setting        | Match / Step.Acc |
> |-------------------------------------|---------------------------|------------------|
> | **MobileIPL (ours)**                | Qwen2-VL-7B               | 69.15            |
> | RL (GRPO)                           | Qwen2-VL-7B| 66.29           |
> | Mobile-Agent-v2           | Qwen3-VL-8B               | 32.5             |
> | Mobile-Agent-v2           | Qwen3-VL-30B-A3B          | 68.6             |
> | Mobile-Agent-v2           | Qwen2.5-VL-72B            | 70.3             |
>
> This shows that our IPL training pipeline can achieve competitive or superior performance with a smaller, open-source model, compared to much larger agent-based framework.
>
>
> [1] GUI-R1: A Generalist R1-Style Vision-Language Action Model for GUI Agents
>
> [2] Mobile-Agent-v2: Mobile Device Operation Assistant with Effective Navigation via Multi-Agent

---

> ### Author Response · Authors · 2025-11-27
> **Response to Reviewer YpsF**
>
> Dear Reviewer YpsF,
>
> Thank you again for taking the time to review our paper. As the rebuttal period is approaching its end, we wanted to kindly check whether our responses have addressed your concerns.
>
> If there is anything that is still unclear or any additional feedback you would like us to address, we would be very happy to provide further clarification.
>
> Best regards,
>
> The paper 23626's Authors

---

### Author Response · Authors · 2025-12-02
**Summary of Rebuttal and Clarifications for the Area Chair**

Dear AC,

Thank you very much for your efforts throughout the review process.
To help reduce your workload, we provide below a concise summary of our rebuttal, in case some reviewers are unable to participate in the discussion.


1. Contributions beyond existing RL and agent-based frameworks (raised by Reviewer YpsF)

   We clarified that prior RL methods for GUI agents (e.g., GUI-R1 / GRPO-style training) typically supervise only the final action and ignore the correctness of intermediate thoughts. MobileIPL instead constructs **thought-level DPO pairs on a CoaT-tree**, providing fine-grained supervision over the reasoning process. For parameterized actions such as CLICK and TYPE, we introduce **smooth, structure-aware rewards** (coordinate distance and text similarity) rather than binary 0/1 signals, which stabilizes training and improves performance. On AITZ, this allows a **7B open-source model** to match or surpass agent frameworks that rely on much larger (30B+) models and closed ecosystems.

2. Relation to TCPO / TreePO / TreeRL / SPO and the choice of IPL (raised by Reviewer 5WvC)

   We added a clearer discussion about recent thought-centric and tree-based methods. TCPO still evaluates thoughts mainly through final actions and does not enforce thought–action consistency in GUI tasks, while TreePO, TreeRL and SPO are designed for **text-only** settings and depend on segmenting long sequences into many segments, which increases cost and reduces data efficiency. In contrast, MobileIPL uses a fixed CoaT-tree tailored to GUI interaction, treats thoughts as priors when generating actions, and reuses multiple samples per thought. After adapting SPO-Chain to our multimodal setting, experiments show that **MobileIPL requires fewer rollouts per sentence while still achieving +1.12% higher accuracy on AITZ**, yielding a better efficiency–performance trade-off than segment-level GRPO.

3. Generalization of action space and hallucinated thoughts (raised by Reviewer 6kud)

   We pointed out that GUI control already involves continuous actions (click coordinates), and our distance-based smooth reward **improves click match on AITZ from 59.72 to 71.45**, suggesting that combining smooth rewards on continuous actions with thought-level preference learning can naturally extend to richer embodied or continuous-control scenarios. We also explained that CoaT-tree sampling with value backpropagation, together with thought-level DPO, explicitly penalizes hallucinated or inconsistent thoughts and pushes the model toward causally useful reasoning instead of merely optimizing final actions.

4. Reward hacking, hyperparameters setting, and the role of instruction evolution (raised by Reviewer Lxqd)

   We discussed that although rule-based or smooth rewards can in principle be exploited, MobileIPL mitigates reward hacking by operating at the thought level, **constructing thought-level supervision signals** and directly optimizing the model’s reasoning process. As for hyperparameters, we reported that a simple configuration (sampling number \(K = 3\), discount factor \(c = 1.0\), and DPO threshold \(= 1/K\)) is both effective and stable, while moving away from it either removes too many informative pairs or introduces many low-contrast pairs that act as noise. Finally, we showed that instruction evolution, evaluated on an auxiliary “instruction → coordinate” task, brings modest gains in greedy accuracy but clear improvements in pass@5, indicating that it mainly **widens the space of potentially correct solutions** and provides the intended diversity prior to IPL.

 Due to the incident that occurred during the rebuttal period, reviewers had very limited time to read and react to our responses before the discussion was frozen. Under normal circumstances, we believe that a full discussion phase would have allowed our rebuttal to directly address the reviewers’ concerns in more depth. We hope this summary helps in assessing the contribution and design choices of MobileIPL.

Sincerely,
The Authors

---

### Author Response · Authors · 2025-12-02
**Summary of Revisions to the Manuscript**

Dear AC and Reviewers,

**We sincerely appreciate your constructive feedback and the time you invested in improving our work.** Following your suggestions, we have revised the manuscript accordingly and marked all the revised content in blue. Below, we summarize the major updates introduced in the revised version:

1. We updated the `related work`, focusing on comparing some recent works similar to ours, such as SPO and TreeRL.
2. We have revised `section 3.3`, adding the definition and implementation details of T-DPO. This content was previously placed in the appendix due to space limitations, which caused misunderstanding.
3. We clarified the definition of d(x,y) and the details regarding the classification of sampling trees to avoid unnecessary misunderstandings.

We thank all reviewers again for helping us improve MobileIPL.

Paper 23626 Authors

---

### Meta-Review · Area_Chair_GarA · 2026-01-07

**Summary:**

1. Major concerns about the technical novelty. Some reviewers suggest this work is more about an engineering combination of existing text-based methods
2. Some concerns about the paper clarity and motivation
3. Questions about the robustness of rule-based rewards as they could be hacked or fail in complex UI layouts.
4. Concerns about the computational cost of the iterative sampling process.

**Reviewer Concerns:**

Authors add more comparison results with the previous baselines, emphasizing the paper novelty and method efficiency. They also review the paper writing. These help address the first and second concerns and mitigate the last concern.

The concerns about possible reward hacking and hallucination remain.

**Reviewer Scores:**

Reviewer YpsF: likely stay the same

Reviewer 5WvC: could be increase with more related work discussions and comparison.

Reviewer 6kud: likely stay the same

Reviewer Lxqd: likely stay the same

---

### Decision · Program_Chairs · 2026-01-26

Accept (Poster)